# SENSOR-INVARIANT TACTILE REPRESENTATION

**Harsh Gupta**[*]  **Yuchen Mo**[*]  **Shengmiao Jin**  **Wenzhen Yuan**
University of Illinois Urbana-Champaign
{hgupt3,yuchenm7}@illinois.edu

## ABSTRACT

High-resolution tactile sensors have become critical for embodied perception and robotic manipulation. However, a key challenge in the field is the lack of transferability between sensors due to design and manufacturing variations, which result in significant differences in tactile signals. This limitation hinders the ability to transfer models or knowledge learned from one sensor to another. To address this, we introduce a novel method to extract Sensor-Invariant Tactile Representations (SITR), enabling zero-shot transfer across optical tactile sensors. Our approach utilizes a transformer-based architecture trained on a diverse dataset of simulated sensor designs, allowing generalizability to new sensors in the real world with minimal calibration. Experimental results demonstrate our method's effectiveness across various tactile sensing applications, facilitating data and model transferability for future advancements in the field.

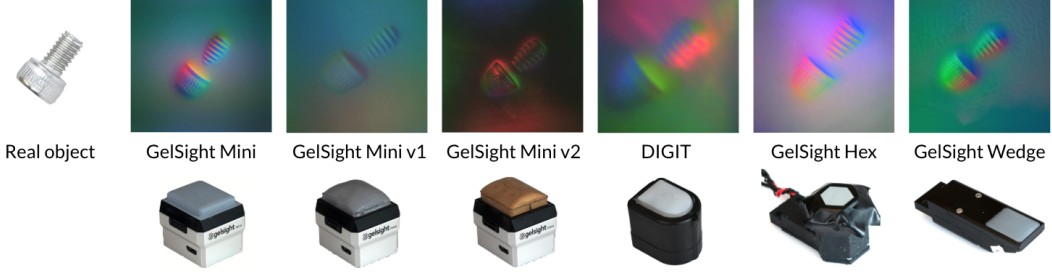

Figure 1: Vision-based tactile sensors vary in both optical design and physical properties. Even with the same contact object, a screw, the tactile images produced by each sensor differ significantly. These variations highlight the challenge of transferring models from one sensor to another.

## 1 INTRODUCTION

Tactile sensing is a crucial modality for intelligent systems to perceive the physical world. Among the various tactile technologies, the GelSight sensor (Yuan et al., 2017a) and its variants (Wang et al., 2021; GelSight, 2024; Zhao & Adelson, 2023) have recently emerged as one of the most influential tactile technologies, offering rich and detailed information on contact surfaces. GelSight captures fine contact geometries through an optical system that transforms tactile data into visual images. This enables robots to precisely detect object shapes, recognize materials, and perform fine-grained manipulations with a high degree of accuracy (Yuan et al., 2018; Dong et al., 2019; Hogan et al., 2020; Ota et al., 2023; Yang et al., 2023; Shirai et al., 2023).

Despite their advantages, GelSight-like sensors, and vision-based tactile sensing in a more general sense, still face a key challenge: sensor variance. Differences in the optical design or manufacturing process can result in significant discrepancies in sensor output. Consequently, machine learning models trained on data from one sensor often fail to generalize to other sensors. This challenge is further compounded by the high cost and effort of collecting tactile datasets, creating a major barrier to sensor transferability in tactile perception.

---

[*] equal contribution

In this paper, we address the challenge of data transferability between GelSight sensors by tackling sensor variance arising from optical design and manufacturing differences. The key issue lies in enabling generalization to new sensors as the domain gap between individual sensors is substantial and unpredictable. Previous methods, such as Yuan et al. (2018); Calandra et al. (2018), attempted to improve generalization by using multiple GelSight sensors to gather diverse tactile datasets, but this approach offered limited gains. More recently, T3 (Zhao et al., 2024) sought to improve transferability by pre-training a transformer model across multiple sensors and tasks. However, their reliance on category-specific encoders limited the ability of their model to generalize to unseen sensors.

In contrast, we propose that achieving sensor transferability requires learning effective sensor-invariant representations by ensuring the model is trained on sufficiently diverse sensor variations. We introduce a novel framework for generating sensor-invariant feature representations from high-resolution tactile readings, enabling zero-shot transfer to unseen sensors across multiple downstream tasks. Our framework incorporates three core innovations:

1. We utilize a small set of easy-to-acquire calibration images to characterize individual sensors. We then use a transformer model as the encoder to effectively combine the calibration images with the tactile reading.

2. We employ supervised contrastive learning (SCL) (Khosla et al., 2020) to emphasize the geometric aspects of tactile data, encouraging the clustering of similar contact geometries across multiple sensors. This training is further supervised by measuring geometric accuracy.

3. We develop a large-scale synthetic dataset using a physics-based simulator that models sensor optical systems, capturing variations in both sensor characteristics and contact geometries. This dataset, consisting of 1M examples across 100 sensor configurations, provides the diversity necessary for robust model training using precise ground truth of the contact geometries.

Our motivation comes from the belief that contact geometry is one of the most critical features for most tactile-driven tasks, including shape recognition, texture classification, and contact localization. By focusing on geometric accuracy and using calibration to remove sensor-specific variations, we ensure the development of robust, sensor-invariant representations. Leveraging physics-based simulations allows us to efficiently generate diverse tactile datasets, reducing the time and cost of real-world data collection.

We evaluate the generalizability of our method across various downstream tasks using multiple real-world GelSight sensors. Our results demonstrate that models trained on one sensor can be seamlessly transferred to others in a zero-shot manner, significantly outperforming existing approaches. This framework paves the way for easier transferability of machine learning models and datasets between different sensors, thereby enhancing the future development of the tactile-sensing community.

## 2 RELATED WORKS

In the realm of vision-based tactile images, the application of computer vision models and algorithms has become common practice due to the visual nature of the data these sensors capture (Dong et al., 2021; Li et al., 2019; Calandra et al., 2018). Researchers have adapted mature representation learning methods from the vision community to tactile images. One popular approach is contrastive learning. Both tactile and visual-tactile representations have been explored for specific tasks (Yuan et al., 2017b; Yang et al., 2022; Tian et al., 2020; Kerr et al., 2022; Guzey et al., 2023; Grill et al., 2020; Zambelli et al., 2021). Another technical approach is based on auto-encoding representation. Cao et al. (2023) and Xu et al. (2024) leveraged Masked Auto-Encoder (MAE) to learn tactile representations.

However, many works that directly apply existing representation learning methods to the tactile modality ignore the significant domain gap seen between sensors. Representations trained on one sensor may work well on the exact same sensor or the same type, but the domain gap between different sensors makes models based on such representation fail to generalize to other sensors. To address this, Zhao et al. (2024) trained individual encoder-decoder pairs for different sensor-task combinations, focusing on learning the shared features and improving fine-tuned performance on

new sensor-task combinations. Yang et al. (2024) sought to address this by proposing a general-purpose multimodal representation for vision-based tactile sensors. By integrating multiple tactile datasets into a large language model (LLM) framework and encoding sensor types as tokens, they try to inform the LLM explicitly of the domain gap among different types of sensors. Higuera et al. (2024) introduced a family of self-supervised models for vision-based tactile sensing that learns general-purpose representations by leveraging masking and self-distillation pretraining across multiple sensors. They aim to improve tactile perception and downstream task performance under a limited labeled data budget. However, these methods often depend on large datasets and treat sensor types as fixed categories, failing to account for variations within the same sensor type and lacking the flexibility to generalize zero-shot to unseen sensors.

Our framework introduces a novel combination of geometry-preserving supervision, supervised contrastive learning, and sensor-specific calibration images. The calibration images capture sensor-specific domain features, such as optical properties unique to each sensor, which help the encoder adapt to these characteristics. By accounting for subtle variations both within the same sensor type and across different types, our method enhances zero-shot generalization across tactile tasks and demonstrates strong transferability to new sensors.

# 3 SENSOR-INVARIANT REPRESENTATION LEARNING

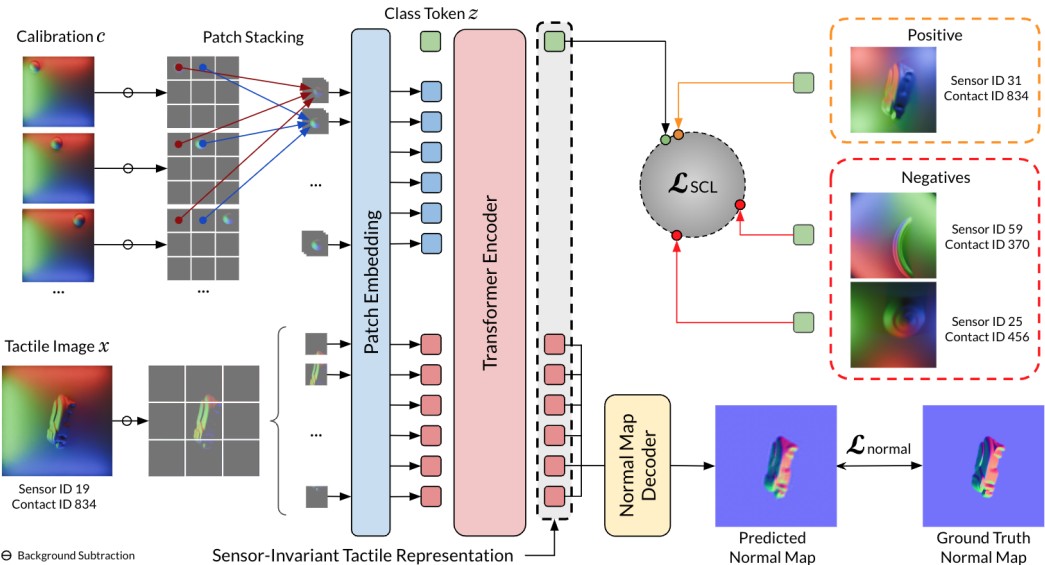

Figure 2: Our sensor-invariant representation learning framework. Each tactile image $x$ is paired with a set of calibration images $c$. We patchify and linearly project $x$ and $c$ to tokens. Additionally, the $c$ patches are region-wise stacked before projection. We concatenate the input tokens with a class token $z$ and pass it through a transformer encoder. The class token $z$ is trained with SCL, while patch tokens are supervised by normal map reconstruction loss. We highlight in grey the concatenation of the output class token and patch tokens as our Sensor-Invariant Tactile Representation (SITR) for downstream tasks.

In this section, we introduce our framework for training Sensor-Invariant Tactile Representation (SITR). We explain how calibration images capture sensor-specific information and use normal maps to preserve contact features. We introduce our implementation of SCL to align tactile features across sensor domains. We provide details on the role of calibration in Section 3.1, followed by the network architecture and training process in Section 3.2.

## 3.1 CALIBRATION IMAGES FOR TACTILE SENSORS

GelSight-like sensors map RGB values at each pixel to the local surface gradient, enabling the reconstruction of the contact surface. However, these sensors exhibit variations in physical properties that

introduce sensor-specific artifacts in tactile images. A widely adopted calibration technique involves pressing a ball of known radius onto the sensor pad at various points. The tactile images captured during this process, combined with the known geometry of the ball, establish the correspondence between RGB changes and the local surface gradient at different locations. This method generates a sensor-specific look-up table. While traditional techniques assume pixel-invariant projection for simplicity, neural networks can further learn precise and pixel-dependent projections.

In the pre-training stage of SITR we adopt these steps to inform the model of sensor characteristics. We include a cube in our calibration to inform SITR about how the gel deforms around edges and corners. Thus, we press two objects—a 4mm diameter ball and a cube corner—at nine locations each, roughly arranged in a $3 \times 3$ grid pattern across the sensor surface as seen in Fig. 3. These calibration images guide the encoder to identify and factor out sensor-specific features.

Formally, given a tactile image $x_i \in \mathbb{R}^{H \times W \times C}$, we select $K$ calibration images $c_{i,k} \in \mathbb{R}^{H \times W \times C}$, where $(H, W)$ is the resolution of the original image and $C$ is the number of channels. To efficiently encode multiple calibration images we reshape $c_{i,k}$ into the form $c_i \in \mathbb{R}^{H \times W \times KC}$, in effect stacking the patches. We then linearly project $x_i$ and $c_i$ into a sequence of $N$ flattened 2D patches $x_p \in \mathbb{R}^{N \times P^2 C}$ and $c_p \in \mathbb{R}^{N \times P^2 C}$ similar to a standard ViT, where $N = HW/P^2$. The resulting token sequences from $x_p$, $c_p$, and a class token $z_i$ are concatenated as input to the transformer encoder.

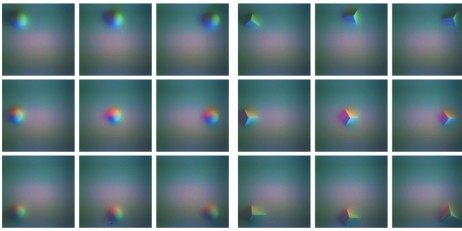

Figure 3: Calibration images used in SITR, obtained by pressing two objects—a 4mm ball and a cube corner—at nine different locations each in a $3 \times 3$ grid.

## 3.2 NETWORK ARCHITECTURE

**Input**: We use the tactile image and a set of calibration images for the sensor as inputs for the network. We subtract the sensor background from all the input images to get the pixel-wise color change as described in Section 3.1. Following the process described in Vision Transformer (ViT) (Dosovitskiy et al., 2021) and Section 3.1, we linearly project the input and calibration images to tokens. Note that calibration images need only be tokenized once per sensor.

**Encoder**: We modify a ViT to process both image and calibration tokens. Adapted from ViT, we add positional encoding to them based on their 2D coordinates and then pass them into the encoder. We apply two supervision signals to train this encoder. One is the pixel-wise normal map reconstruction loss for the output patch tokens, and an additional contrastive loss for the class token.

**Normal map reconstruction**: During the SITR pre-training phase, we apply a lightweight decoder to reconstruct the contact surface as a normal map from the encoder output. Normal maps record the orientation of each 3D point on the contact surface. This feature is invariant to the variance across different sensors, contains rich geometry information for downstream tasks, and is viable for many GelSight-like vision-based tactile sensors. Therefore, we apply a pixel-wise MSE loss $\mathcal{L}_{normal}$ between predicted normal map $\hat{n}$ and ground truth normal map $n$.

**Supervised contrastive learning**: SCL is an extension of contrastive learning that leverages label information to learn more effective representations. Traditional contrastive learning aims to pull together similar samples and push apart dissimilar ones in the embedding space, typically relying on data augmentations to create positive pairs. SCL enhances this approach by utilizing class labels to define similarity, allowing for more semantically meaningful contrasts.

We employ SCL to create sensor-invariant representations from our labeled simulated tactile dataset. We label positive pairs from tactile images with the same contact geometry across multiple sensors, while negative pairs are labeled from images of different contact geometries or locations. In our batched implementation, we include two views for each sample: tactile images of the same contact captured by two different sensors. This approach allows us to learn discriminative features for downstream tasks while being robust to variations in sensor characteristics.

Formally, given a batch of $N$ samples, let class token $z_i \in \mathbb{R}^d$ represent the encoded feature vector for sample $i$, where $d$ is the dimension of the embedding space. Let $y_i$ denote its corresponding

contact label. Let $A(i)$ denote the set of all samples in the batch except for sample $i$ itself. For each anchor sample $i$, we define the set of positive samples as $P(i) = p \in A(i) : y_p = y_i$, with $|P(i)|$ being its cardinality. The supervised contrastive loss for a batch of samples is then formulated as

$$\mathcal{L}_{\text{SCL}} = \sum_{i \in I} \frac{-1}{|P(i)|} \sum_{p \in P(i)} \log \frac{\exp\left(\frac{z_i \cdot z_p}{\tau}\right)}{\sum_{a \in A(i)} \exp\left(\frac{z_i \cdot z_a}{\tau}\right)}$$

where $\tau$ is a temperature parameter that scales the similarity values to control the concentration of the distribution.

In summary, the total loss for SITR is defined as $\mathcal{L} = \lambda_{\text{normal}} \cdot \mathcal{L}_{\text{normal}} + \lambda_{\text{SCL}} \cdot \mathcal{L}_{\text{SCL}}$ where $\lambda_{\text{normal}}$ and $\lambda_{\text{SCL}}$ are loss weighting hyperparameters. Refer to Section A.1 for more implementation details.

## 4 DATASETS

We collect three datasets for model training and evaluation. The first dataset contains purely synthetic data and is used to train the encoder for SITR. The other two datasets are collected across 7 real sensors on two specific tactile applications: object classification and contact localization. These two datasets are used to evaluate the zero-shot transferability of SITR for downstream tasks.

### 4.1 SIMULATED TACTILE DATASET

We construct a large-scale synthetic dataset that spans a wide range of tactile sensor configurations, providing tactile signals of contact geometries along with their corresponding normal maps. The sensor's configuration is defined by its optical design, such as the location and optical properties of the lights, cameras, and reflective surfaces. These attributes quantify the major variances seen in real tactile sensors. The core idea is to train SITR with a large distribution of simulated sensors so SITR can generalize to, and be aligned across, real-world sensors. This dataset is designed to be sensor-aligned, where each contact geometry is sampled across all sensor configurations for SCL.

We use Physics-based Rendering (PBR) (Pharr et al., 2023) to simulate GelSight sensors (Agarwal et al., 2021) and implement the algorithm in Blender. PBR simulates the camera images by tracing the path of light rays traveling in the scene and how they interact with optical components. Therefore, the technology models the physical behavior of the optical system and can simulate a GelSight sensor's reading with parameterized optical settings. We design the simulator platform to customize the sensors by modulating the locations and characteristics of each optical component. Fig. 4 illustrates an example setup where three light sources surround a deformable surface. A camera positioned above captures the change of color on the surface caused by object contact.

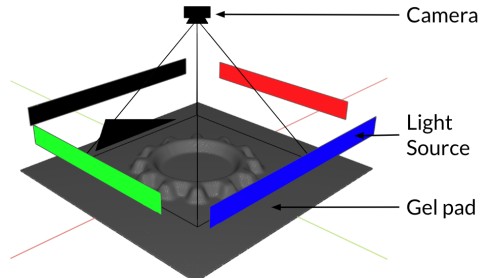

Figure 4: Demonstration of our physics-based rendering (PBR) model to simulate GelSight sensors. We parameterize the sensor's optical design in the environment.

**Sensor variation**: To mimic the variance across real-world tactile sensors, we identify key parameters that highlight the differences between real-world tactile sensors. Specifically, we look at the differences among GelSight Mini (GelSight, 2024), GelSight Hex (Yuan et al., 2017a), GelSight Wedge (Wang et al., 2021), GelSlim 3.0 (Taylor et al., 2022), GelSight Finray (Liu & Adelson, 2022), and DIGIT (Lambeta et al., 2020). This includes light properties (shape, orientation, angle, color), gel properties (stiffness, specularity), and camera properties (FOV, sensing area). In total, we generate 100 unique simulated sensor configurations. More details on the sensor configurations and examples of rendered images for the same contact object can be found in Section A.2. For each sensor, we also collect a set of calibration images as described in Section 3.1. We introduce random variability in the calibration positions to make the training more robust to the real-world setting.

**Object diversity**: To enable SITR to generalize across diverse contact geometries, we utilize 50 high-resolution 3D meshes of common household objects. These meshes include tools, kitchenware, toys, and clothing items, which are often used in robotics research. During simulation, the objects are randomly scaled, rotated, and placed at varying locations on the gel pad. For each contact geometry, we render tactile images using all sensor configurations and pair them with ground-truth surface normal maps. We generate a total of 10K contact configurations through this process.

With 10K unique contact configurations across 100 different sensor configurations, we pre-train SITR encoder solely using our 1M synthetic dataset.

## 4.2 REAL-WORLD TACTILE DATASET

We collect real-world datasets for training and evaluating downstream tasks across different baselines. Compared to the synthetic dataset we used for SITR encoder pre-training phase, we keep SITR encoder frozen and train only the corresponding task-specific decoder head for downstream tasks. We use seven different sensors for our datasets: four GelSight Minis (GelSight, 2024) with varying sensor bodies and in-house gel pad modifications, GelSight Hex (Yuan et al., 2017a), Gel-Sight Wedge (Wang et al., 2021), and DIGIT (Lambeta et al., 2020).

For the classification task, we select 16 objects and press them against the sensor in various poses and depths, recording 1K tactile images for each object. We repeat this process for all 16 objects across the 7 sensors, resulting in a dataset with 112K tactile images, with 16K samples per sensor. Section A.3 shows that tactile signals vary even when using the same object across different sensor configurations.

For the pose estimation task, we modify an Ender-3 Pro 3D printer by replacing its extruder with 3D-printed indenters and mount the tactile sensors onto the print bed. This setup provides the accurate ground truth pose of each contact, including metric $x$, $y$, and $z$ values. During the data collection process, we press indentors at various locations and depths on the sensor surface. We collected 1K samples per indentor for 6 different indentors across 4 sensors. This results in a dataset of 24K tactile images with precise pose labels, with 6K samples per sensor. More details can be found in Section A.4.

## 5 EXPERIMENTS

In this section, we show several experiments to evaluate the zero-shot transferability of our model to different real sensors. We evaluate model performance on three downstream tasks: shape reconstruction, object classification, and contact localization.

## 5.1 EXPERIMENT SETTING

We conduct experiments with multiple real tactile sensors that can be divided into two groups:

- Intra-sensor set: GelSight Mini 1 to 4 of different gel pads. These sensors have the same optical design, i.e., placement of camera and light sources, but differ in brightness and color of tactile signals due to manufacturing differences and choice of coating materials.
- Inter-sensor set: GelSight Mini 1, GelSight Wedge, GelSight Hex, and DIGIT. These sensors are designed with very different optical structures and, therefore, generate tactile signals that are significantly different from each other.

For each downstream task, we freeze the SITR encoder and only train the downstream task-specific decoder on a single sensor. We evaluate this model using the rest of the sensors in the set. Formally, let $S = \{S_1, S_2, \ldots, S_n\}$ be the set of sensors. Let $A_{ij}$ represent the performance (e.g., classification accuracy or pose estimation error) when trained on $S_i$ and evaluated on $S_j$. The transfer performance across all sensors in the set is computed as

$$\text{Transfer Performance} = \frac{1}{n(n-1)} \sum_{i=1}^{n} \sum_{\substack{j=1 \\ j \neq i}}^{n} A_{ij}$$

We also compute the score when training and testing on the same sensor $i = j$ acting as an upper bound of the performance: No Transfer Performance $= \frac{1}{n} \sum_{i=1}^{n} A_{ii}$.

**Baseline**: We compare our SITR with ViTs that are either trained from scratch or fine-tuned from ImageNet weights to show the effectiveness of our method. As there is no previous work that directly focuses on transferable tactile representations, we also compare against T3 (Zhao et al., 2024) and UniT (Xu et al., 2024). T3 focuses on improving few-shot fine-tuning results across different sensors and has the potential for zero-shot transfer. UniT learns dense representations for various downstream tasks and shows preliminary results on transferring among GelSight Mini sensors. We evaluate their available models for our experiments to compare the transferability of these representations. Additionally, we ablate the method used in T3 (MAE) and UniT (VQGAN) when trained on our synthetic dataset to test the effectiveness of our architecture in Section A.6. We describe model configurations and decoders for each task in Section A.1.

## 5.2 ZERO-SHOT TRANSFER FOR SHAPE RECONSTRUCTION

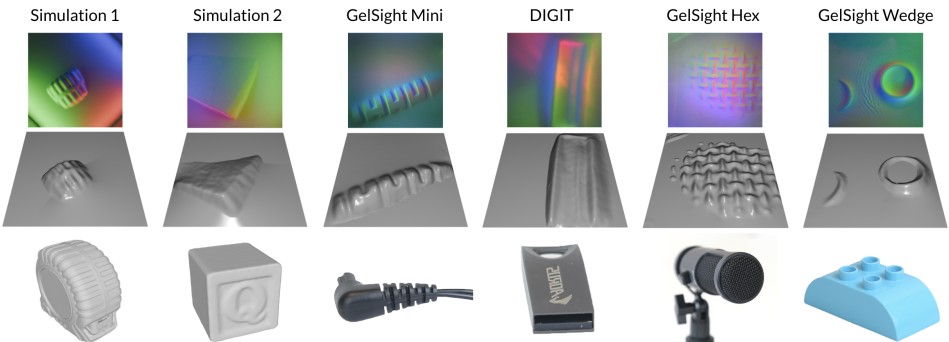

Figure 5: Reconstruction examples for various sensors. The top row shows input tactile images, the middle row presents 3D reconstructions, and the bottom row shows the contact objects. Simulated sensors (Simulation 1 and 2) are in the training set, while real sensors (GelSight Mini, DIGIT, Hex, Wedge) are not.

We qualitatively evaluate how SITR preserves geometry and texture information by reconstructing the contact height map. As shown in Fig. 5, we reconstruct normal maps for objects in our real-world classification dataset and integrate them to generate their corresponding height maps. These 3D reconstructions capture fine-grained geometry and texture details of the contact surface. Though, these reconstructions are naturally constrained by the resolution and sensitivity limitations of the sensors. Despite these limitations, the preservation of dense surface features demonstrates the robustness of SITR in accurately modeling the contact geometry across varying sensor inputs.

## 5.3 OBJECT CLASSIFICATION

We compare SITR with baselines using our real-world classification dataset from Section 4.2 and report top-1 accuracy. We freeze our SITR encoder and train the downstream classifier using cross-entropy loss. For T3, we use their released GelSight Mini encoder weights for intra-sensor experiments. Since T3 does not provide encoder weights for GelSight Hex or DIGIT, we report inter-sensor results only for the GelSight Wedge and Mini. Note that T3's encoders were trained on marked sensors, so the results in our unmarked evaluations may not reflect their full potential. UniT demonstrates transferability only within GelSight Minis, so we exclude it from inter-sensor experiments. We train a UniT encoder on our unmarked real-world dataset and evaluate its intra-sensor transfer performance.

As shown in Table 1, SITR outperforms all baselines by a large margin regarding classification accuracy when transferred across sensors. Note that most models perform well under the no-transfer setting, but fail to generalize when tested on a different sensor. This indicates that baselines can understand tactile features learned in the same domain, but SITR can capture meaningful features that are robust to changes in the sensor domain. We also find that the ViT pre-trained on ImageNet

| Method | Intra-sensor set ↑ | Inter-sensor set ↑ | Wedge-Mini ↑ | No transfer ↑ |
|---|---|---|---|---|
| ViT-Base Scratch | $36.90 \pm 22.19$ | $24.02 \pm 14.83$ | $52.56 \pm 4.95$ | $96.76 \pm 1.41$ |
| ViT-Base Pre-trained | $73.22 \pm 22.42$ | $48.10 \pm 22.82$ | $76.28 \pm 17.06$ | $99.01 \pm 1.14$ |
| ViT-Large Pre-trained | $78.38 \pm 17.79$ | $54.34 \pm 23.04$ | $79.04 \pm 16.44$ | $99.44 \pm 0.43$ |
| T3-Medium | $38.66 \pm 20.63$ | $--$ | $17.02 \pm 8.55$ | $93.77 \pm 2.87$ |
| UniT | $46.39 \pm 23.30$ | $--$ | $--$ | $92.53 \pm 4.19$ |
| SITR (Ours) | $\mathbf{90.23} \pm 8.16$ | $\mathbf{81.94} \pm 12.92$ | $\mathbf{90.80} \pm 2.85$ | $\mathbf{99.72} \pm 0.22$ |

Table 1: Results of object classification accuracy on 16 classes for model transfer and no-transfer performance. We report the mean and standard deviation of transfer accuracy percent among the sensor sets specified. Random guess classification accuracy corresponds to 6.67%.

performs better than that trained from scratch, which indicates the effectiveness of pre-training on the image domain.

Moving to feature-level analysis, Fig. 6 presents the t-SNE visualization of the SITR features for the contacts in our real-world classification dataset. The visualization illustrates that the use of contrastive learning significantly improves feature clustering, bringing together samples of the same object across different sensors. This indicates that SITR successfully aligns the tactile signals from different sensors, highlighting its capacity to eliminate sensor-variant features.

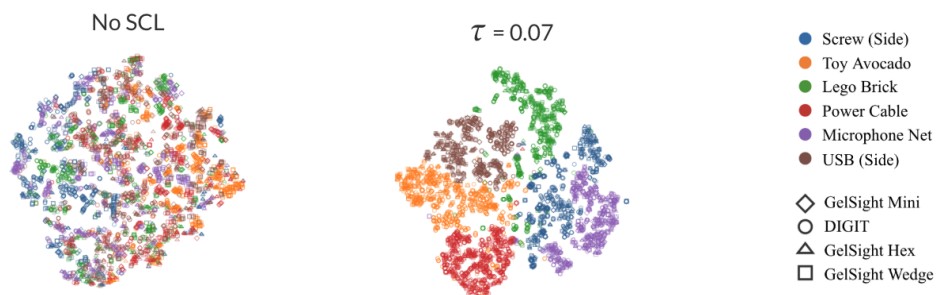

Figure 6: t-SNE visualization of the feature space. We qualitatively show that our contrastive loss term helps cluster those similar contacts from different sensors together.

However, the results also reveal some challenges. The features from the DIGIT sensor are somewhat more difficult to cluster with those from other sensors. This is better demonstrated in our detailed transfer results in Sec. A.5.1, where we see relatively worse classification transferability to and from the DIGIT sensor. We attribute this to DIGIT's distinct optical design, which differs from the GelSight designs in our simulation dataset. We believe the result will be improved in the future if we extend our synthetic dataset to cover optical designs similar to the DIGIT sensor.

## 5.4 POSE ESTIMATION

In this task, we try to estimate the 3-DoF $(x, y, z)$ position change of the object in contact using an initial and final tactile image. We separately feed 2 tactile images of the same object into the frozen SITR encoder, concatenate their features, and train a decoder to learn the pose change with mean square error (MSE) loss. For baseline models, we use similar pipelines as detailed in Section A.1. We evaluate this task on the inter-sensor set to see how each model handles differences in scale across sensors. Each sensor in this set has a different physical design, meaning they capture tactile signals at varying scales. Variations in object size may create significant challenges for zero-shot transfer tasks like pose estimation.

As shown in Table 2, SITR demonstrates strong performance on the pose estimation when tested on a different sensor, reducing the RMSE by about 50% compared to baselines. Remarkably, all models have similar RMSE errors for the no-transfer setting. This may suggest sub-millimeter inaccuracies present in our data collection process. Nonetheless, the no-transfer setting serves as the upper bound

| Method | Inter-sensor set ↓ | Wedge-Mini ↓ | No transfer ↓ |
|---|---|---|---|
| ViT-Base Scratch | $1.63 \pm 0.20$ | $1.69 \pm 0.13$ | $0.56 \pm 0.02$ |
| ViT-Base Pre-trained | $1.58 \pm 0.22$ | $1.65 \pm 0.13$ | $\mathbf{0.49} \pm 0.01$ |
| ViT-Large Pre-trained | $1.49 \pm 0.25$ | $1.45 \pm 0.01$ | $0.50 \pm 0.02$ |
| T3-Medium | $--$ | $1.7 \pm 0.07$ | $0.51 \pm 0.02$ |
| SITR (Ours) | $\mathbf{0.80} \pm 0.21$ | $\mathbf{0.62} \pm 0.11$ | $0.51 \pm 0.01$ |

Table 2: Results of pose estimation with 6 objects. We report the mean and standard deviation of transfer pose estimation root mean square error (RMSE) in $mm$ among the sensor sets specified. Random guess pose estimation RMSE corresponds to $2.52mm$.

for our transfer setting. We also find that compared to ViT trained from scratch, the ViT pre-trained on ImageNet only marginally improves this task. This indicates that features learned from natural images may not transfer adequately to the tactile domain for accurate regression tasks like pose estimation.

# 6 ABLATIONS

## 6.1 NUMBER AND TYPE OF CALIBRATION IMAGES

**Effect of number of calibration images on**

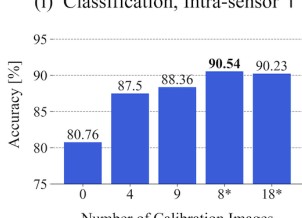
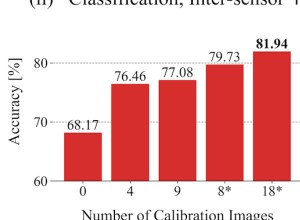
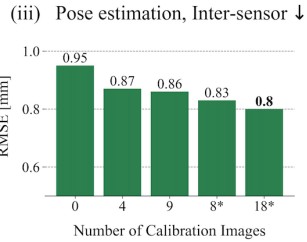

Figure 7: Ablation study on the number and type of calibration images used in SITR, showing their effect on (i) Classification accuracy for inter-sensor transfer, (ii) Classification accuracy for intra-sensor transfer, and (iii) Pose estimation error for inter-sensor transfer.

We conduct an ablation study to investigate the impact of the number and type of calibration images on the performance of SITR. In the standard SITR setup, we press two objects—a ball and a cube corner—at nine locations roughly arranged in a 3x3 grid pattern across the sensor surface. To explore variations, we retrained SITR using different subsets of these calibration images and evaluated performance across all downstream tasks.

We test on five calibration configurations: No calibration images (0); Ball pressed at 4 corners (4); Ball pressed in a 3x3 grid (9); Ball and cube pressed at 4 corners (8*); Ball and cube pressed in a 3x3 grid, which is the standard setup (18*). Fig. 7 illustrates how different numbers and types of calibration images impact SITR's performance. We observe that increasing the number of calibration images increases performance across all tasks. However, the performance gains diminish as more images of the same object are added (as seen in the progression from cases (0) to (4) to (9)). Introducing a second calibration object with a distinct geometry, such as the cube (cases (4) to (8*)), results in a larger performance boost compared to simply adding more images of the same object (cases (4) to (9)). The effect of calibration images is particularly notable in the inter-sensor setting, where we see upwards of a 20% increase in classification accuracy from case (0) to (18*). We choose case (18*) for SITR since increasing the number of calibration images does not incur additional inference costs, as calibration tokens are computed only once per sensor.

## 6.2 CONTRASTIVE LOSS AND TEMPERATURE

We conduct an ablation study to assess the effect of SCL and varying contrastive temperatures $\tau$ on SITR's performance. Specifically, we compared models with and without the SCL term and tested

**Effect of contrastive learning temperature on**

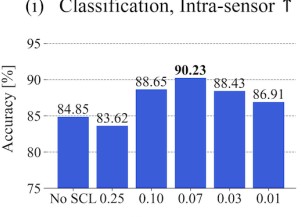

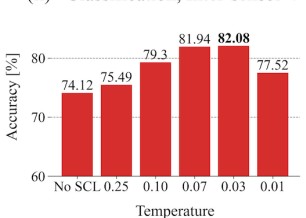

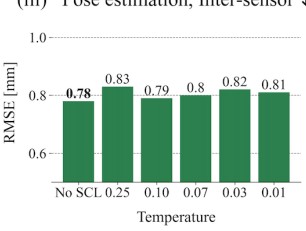

Figure 8: Ablation study examining the impact of SCL and varying contrastive temperature $\tau$ on SITR's performance. Subplots (i) and (ii) show classification accuracy in inter-sensor and intra-sensor settings, respectively, while (iii) shows the effect on pose estimation RMSE.

five contrastive temperatures: 0.25, 0.10, 0.07, 0.03, and 0.01. No SCL corresponds to using only the normal map reconstruction loss during pre-training. Results in Fig. 8 show that a contrastive temperature of 0.07 achieves the best classification performance in the intra-sensor setting, while 0.03 performs best for the inter-sensor setting. Lower or higher temperatures lead to reduced performance in both cases. For the pose estimation task, the addition of SCL has a negligible impact on the RMSE. These results suggest that contrastive learning helps align features across sensors in classification tasks. However, in the pose estimation task, the model's performance is more dependent on the fine-grained geometry information from the contact surface. For SITR, we choose a temperature of 0.07 for its strong performance in the classification task.

# 7 DISCUSSION

Our qualitative and quantitative results indicate that SITR can generalize across sensors while preserving key geometric and texture features from tactile interactions. Our model has been largely trained and evaluated on optical tactile sensors with flat gel pads within the GelSight family. Despite this, SITR can be adapted to a broader range of sensors. Our PBR environment can be easily expanded to accommodate new parameters to explore distinct optical properties in flat tactile sensors. For more complex optical sensors like GelSight Svelte (Zhao & Adelson, 2023) or DIGIT 360 (Lambeta et al., 2024), adaptation remains feasible using an appropriate PBR model and contact surface mapping.

One future direction of our framework is to generalize to traditional array-based tactile sensors. The challenge lies in bridging the signal modalities of low-resolution normal force distribution to the high-resolution contact geometry from GelSight sensors. One possible approach is to downsample vision-based tactile sensors' depth maps to approximate low-resolution tactile signals while establishing a meaningful invariant relationship between depth and force. While this approach provides a step towards a unified tactile modality, its effectiveness in maintaining transferability requires further validation and exploration.

Another direction of future work is incorporating marker-based tactile information to SITR. Many variations of GelSight are equipped with markers—distinct patterns embedded within the gel surface—that provide force and torque information. Currently, these markers are reconstructed using simple computer vision techniques to generate a marker motion field. We believe that unifying marker motion fields between sensors may be possible with adaptations to calibration in SITR. This extension would broaden the applicability of our model to a wider range of tactile sensing tasks.

# 8 CONCLUSION

In this paper, we introduced SITR, a tactile representation that transfers across various vision-based tactile sensors in a zero-shot manner. We build large-scale, sensor-aligned datasets using synthetic and real-world data, and propose a method to train SITR to capture dense, sensor-invariant features. Our experimental results demonstrate that SITR outperforms baseline models and other related tactile representations in different downstream tasks, showcasing robust transferability and effectiveness. SITR represents a step towards a unified approach to tactile sensing, where models can generalize seamlessly across different sensor types, facilitating advancements in robotic and tactile research.

ACKNOWLEDGMENTS

The authors would like to thank Amin Mirzaee and Ruohan Zhang for their help with hardware design and prototyping. The authors thank Ruihan Gao for her assistance with revising the paper. We thank Arpit Agarwal for his insightful discussions on tactile simulation. This work was supported in part by the NSF award #2024882, the Indira Gunda Saladi Engineering Research Prize, and the Illinois Office of Undergraduate Research.

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

## A APPENDIX

### A.1 IMPLEMENTATION DETAILS

This section outlines the detailed implementation steps, including pre-processing, architecture, training settings, and decoder choices for all models.

#### A.1.1 PRE-PROCESSING

For SITR, we apply the following pre-processing steps across real and simulated sensors:

1. All input images are resized to $224 \times 224$. For the GelSight Wedge sensor, an affine transformation is applied to correct distortions in the tactile images.

2. Batched data augmentations are applied during training to both the tactile input and calibration images, including color jitter and Gaussian blur.

3. Background subtraction is performed on each image to isolate the tactile signal. All images are then normalized based on the mean and standard deviation calculated from the simulated dataset.

#### A.1.2 ARCHITECTURE

**Encoders:** Table 3 shows the number of parameters used in each encoder.

| Model | Number of Parameters |
|---|---|
| ViT-Base | 86M |
| ViT-Large | 307M |
| T3-Medium | 173M |
| UniT | 25M |
| SITR (Ours) | 96M |

Table 3: Comparison of model parameters.

Our SITR model is derived from the ViT-Base architecture. The key modification is in the patch embedding, where we tokenize the tactile input and calibration images separately and add a positional embedding before passing them through the transformer.

**SITR Training Decoders:** During the pre-training phase for SITR, we use two decoders:

- **Normal Map Reconstruction Decoder:** We apply a simple linear projection to the output tactile image tokens from SITR. We reshape and unpatchify the output to create a feature image map. We supervise with MSE loss $\lambda_{\mathrm{normal}}$ against the ground truth normal map.

- **Class Token Decoder:** The class token is passed through a linear projection to a 128-dimensional embedding. We then supervise this embedding with SCL loss $\lambda_{\mathrm{SCL}}$.

- **Loss Terms** The total loss during training is a weighted sum of these two loss terms: $\mathcal{L} = \lambda_{\mathrm{normal}} \cdot \mathcal{L}_{\mathrm{normal}} + \lambda_{\mathrm{SCL}} \cdot \mathcal{L}_{\mathrm{SCL}}$ We set both loss weighting hyperparameters $\lambda_{\mathrm{normal}}$ and $\lambda_{\mathrm{SCL}}$ to 1.

**Downstream Task Decoders:** We try several decoders for downstream tasks for each baseline and task and report the best-performing ones here.

1. **Classification Decoders** We use Cross Entropy Loss for this task.
   - **SITR:** We unpatchify the output tokens $x_i$ to a feature map and pass it through a ResNet-18 network. The resulting feature vector is concatenated with the class token $z_i$. We then apply a 3-layer MLP decoder with dimensions [256, 128, 16]. The SITR encoder is frozen during this process.

- **ViT:** For all ViT encoders, we linearly project the class token to an output of 16 dimensions. We also find that unfreezing the ViT pre-trained weights during training improves performance.
- **T3:** We unpatchify the output tokens to a feature map and pass it through a ResNet-18 network with an output dimension of 16. The T3 encoder is frozen for this process.
- **UniT:** We directly apply their proposed pooling and MLP decoder blocks to an output dimension of 16. We find that unfreezing the UniT encoder provides better results.

2. **Pose Estimation Decoders** We use MSE loss for this task.

- **SITR:** We pass 2 tactile images $x_1$ and $x_2$ into the network separately. We unpatchify the output tokens from $x_1$ and $x_2$ and concatenate their feature maps. We pass the concatenated feature maps into a modified ResNet-18 with a 6-channel input. We then linearly project the resulting feature vector to an output dimension of 3. The SITR encoder is frozen during this process.
- **ViT:** For all ViT encoders, we pass 2 tactile images $x_1$ and $x_2$ into a modified ViT network allowing 6 channel input. We then linearly project the resulting class token to an output dimension of 3. We unfreeze the ViTs when training.
- **T3:** We follow the same procedure described in SITR's pose estimation decoder. 2 tactile images $x_1$ and $x_2$ are passed into the network separately. We unpatchify the output tokens from $x_1$ and $x_2$ and concatenate their feature maps. We pass this feature into a modified ResNet-18 and linearly project the resulting feature vector to an output dimension of 3. We keep the T3 encoder frozen during this training process.

## A.2  SIMULATED DATASET

| Parameter | Lower bound | Upper bound | Lower bound vis. | Upper bound vis. | Lower bound env. | Upper bound env. |
|---|---|---|---|---|---|---|
| Light shape | point | area | | | | |
| Light orientation | sides | corners | | | | |
| Light angle | 5° | 30° | | | | |
| Light color | rand | rand | | | | |
| Gel stiffness | low | high | | | | |
| Gel specularity | low | high | | | | |
| Camera FOV | 40° | 90° | | | | |
| Sensing area | 4cm² | 16cm² | | | | |

Table 4: Visualization of parameter bounds in the simulated dataset.

As discussed in Section 4.1, we construct a large-scale simulated dataset that includes a wide range of tactile sensor configurations. Figure 9 illustrates a sample of tactile images from different simulated sensor configurations and contact geometry within the dataset. The samples can be retrieved from our dataset with the sensor IDs and contact IDs provided.

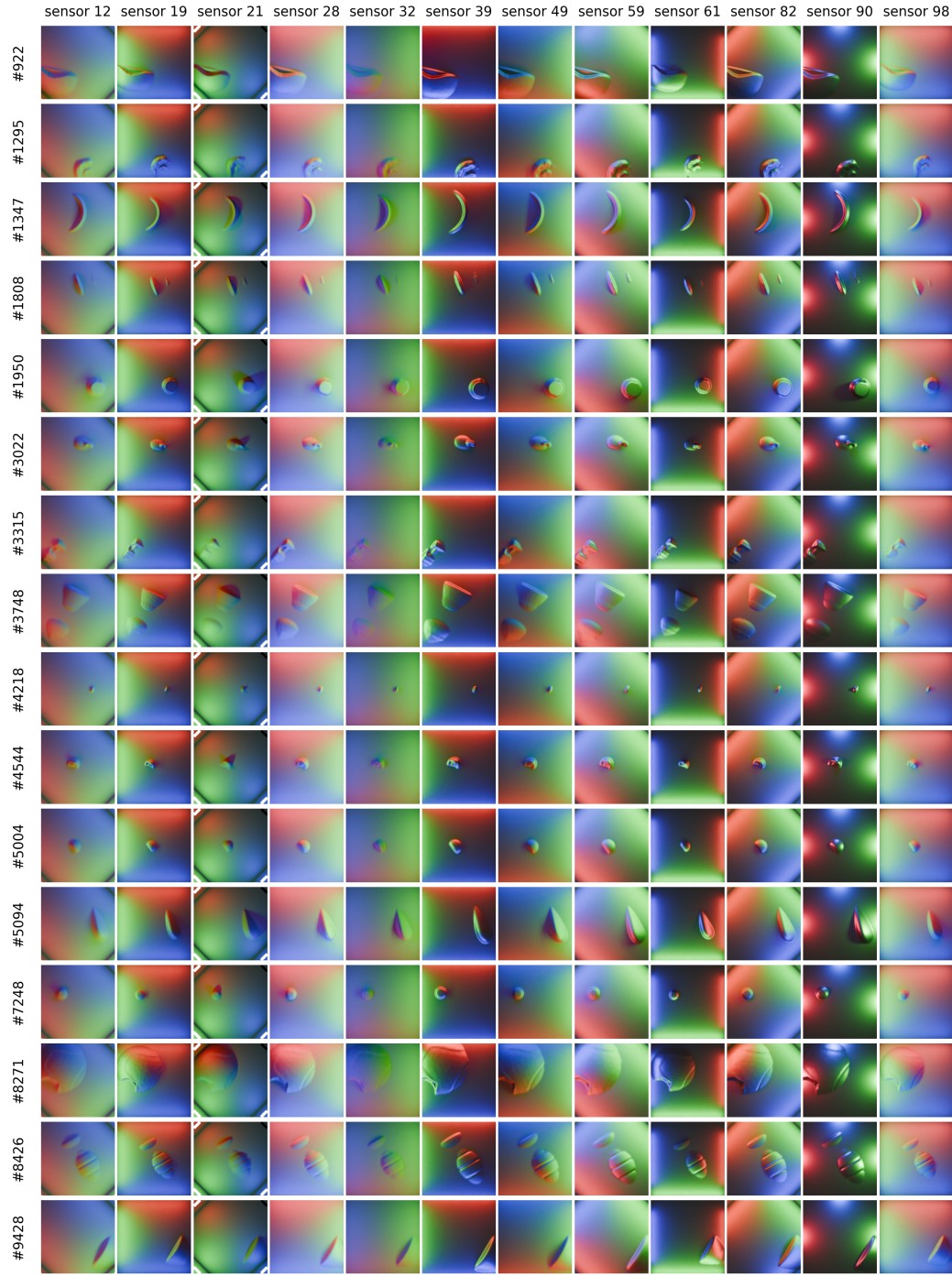

Figure 9: Samples from the simulation dataset.

### A.3 CLASSIFICATION DATASET SAMPLES

Figure 10 and 11 show the real-world classification dataset that we used to generate the result discussed in Section 5.3. Each row corresponds to a different object class, and each column represents a different sensor.

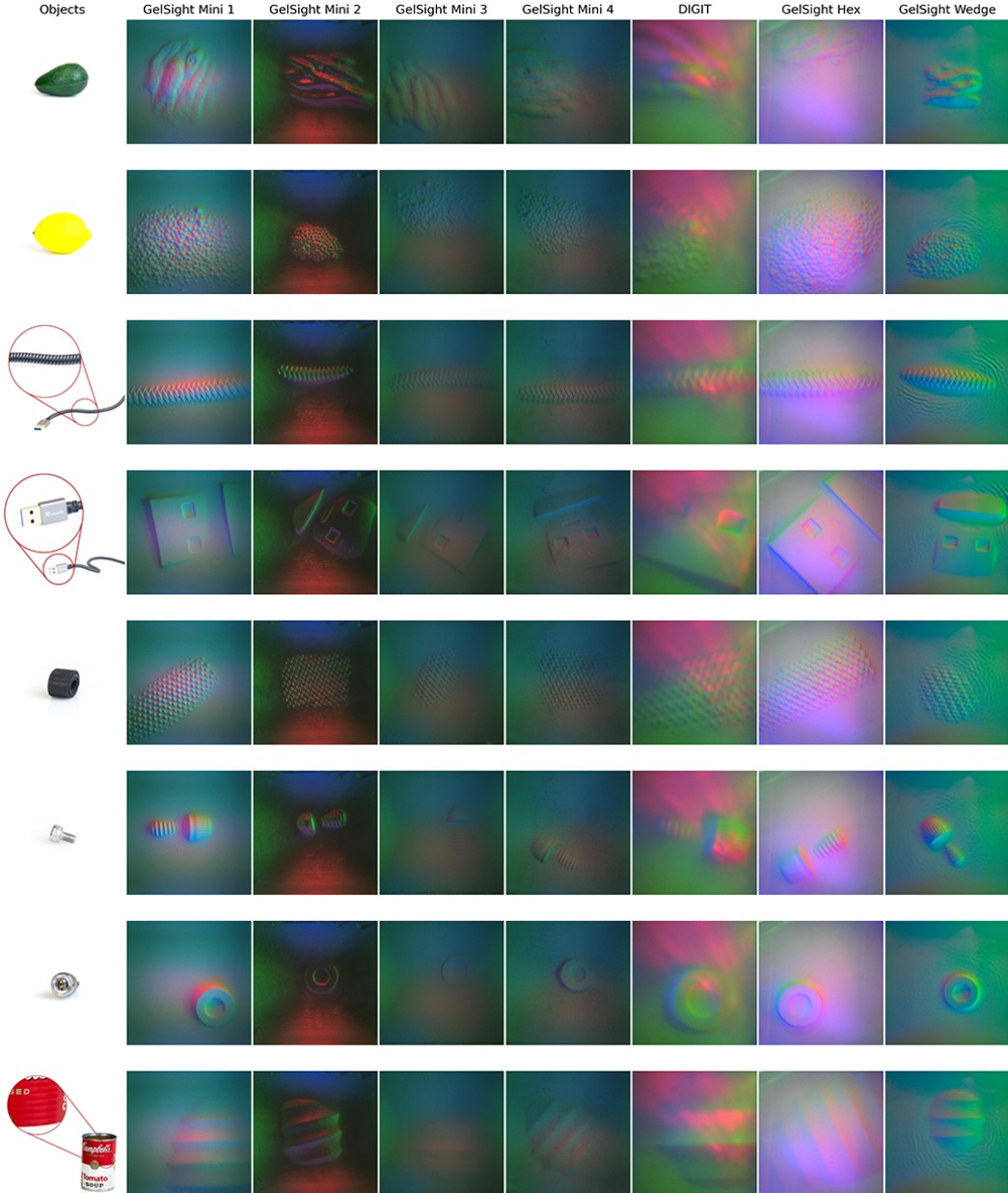

Figure 10: Samples from the classification dataset. (Part 1)

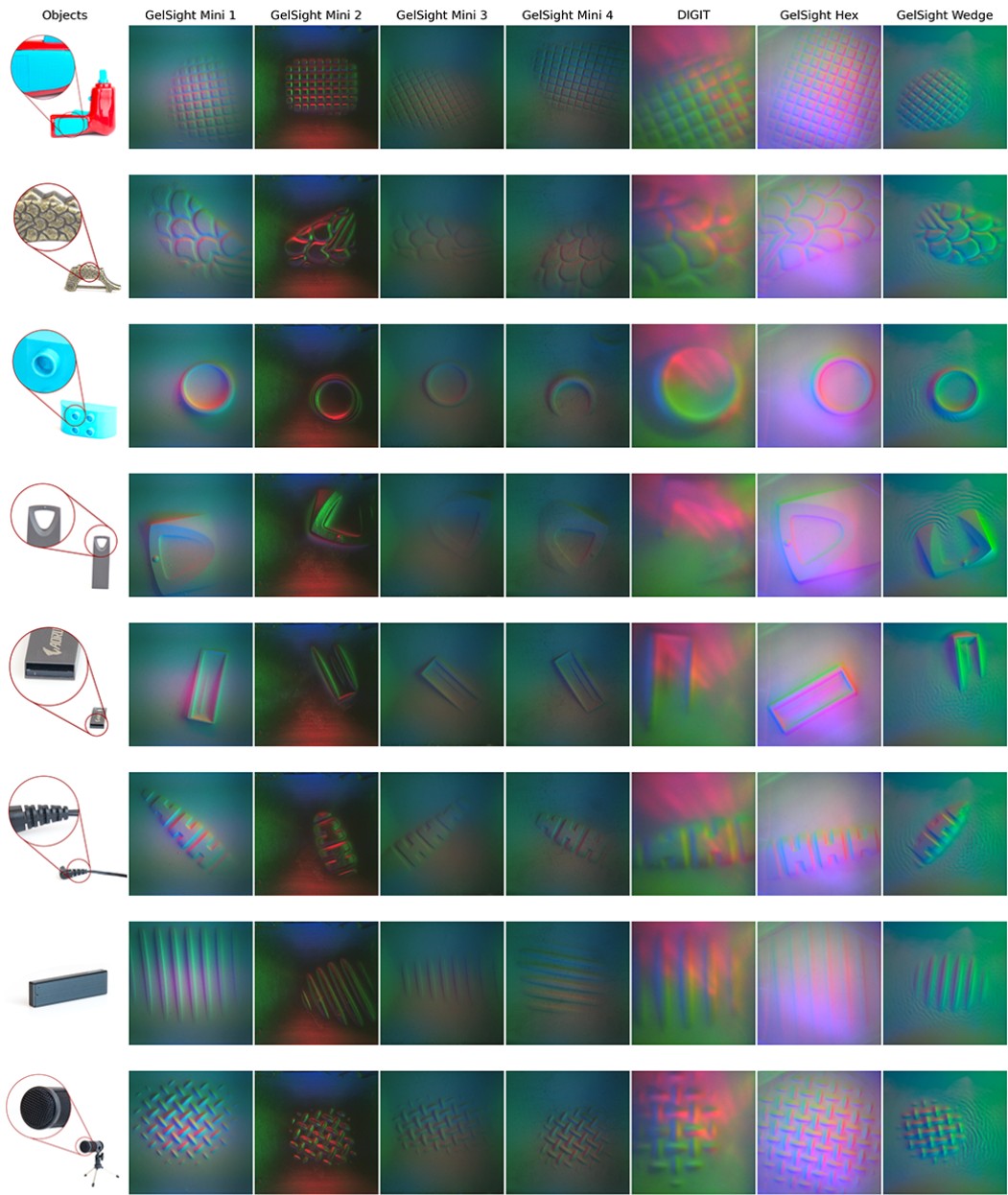

Figure 11: Samples from the classification dataset. (Part 2)

### A.4 POSE ESTIMATION DATASET SAMPLES

Figure 12 shows the real-world classification dataset that we used to generate the result discussed in Section 5.4. Each row corresponds to a different object class, and each column represents a different sensor.

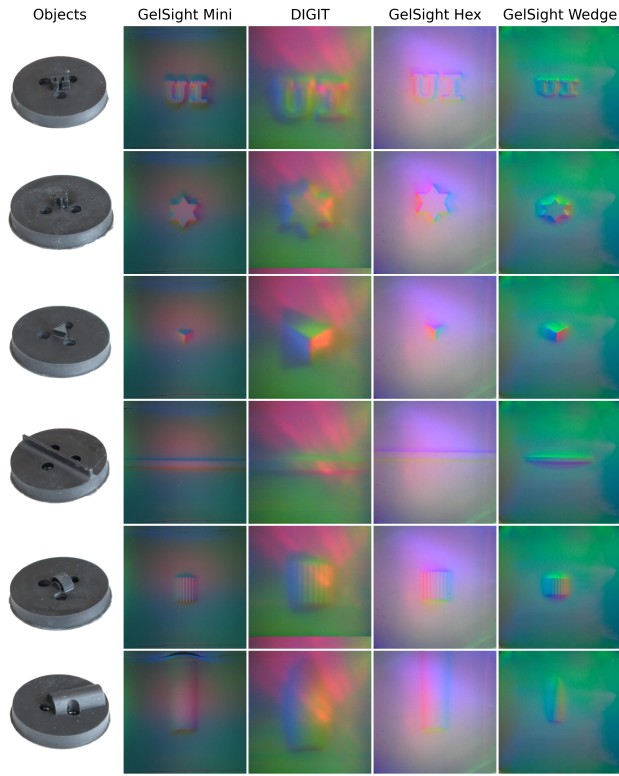

Figure 12: Samples from pose estimation dataset.

Figure 13 shows the modified Ender-3 3D printer. We mount indentors and collect the pose estimation dataset for multiple sensors.

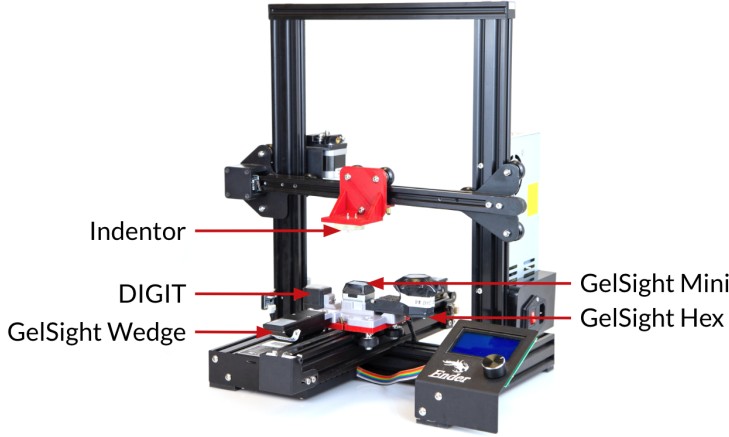

Figure 13: Modified Ender-3 Pro 3D printer

## A.5 TRANSFERABILITY DETAILS

In this section, we present the full results of the sensor transfer downstream experiments. Details of experiments can be found in Section 5.3 and 5.4. Figure 14 and 15 show the classification results and Figure 16 shows pose estimation results.

### A.5.1 CLASSIFICATION

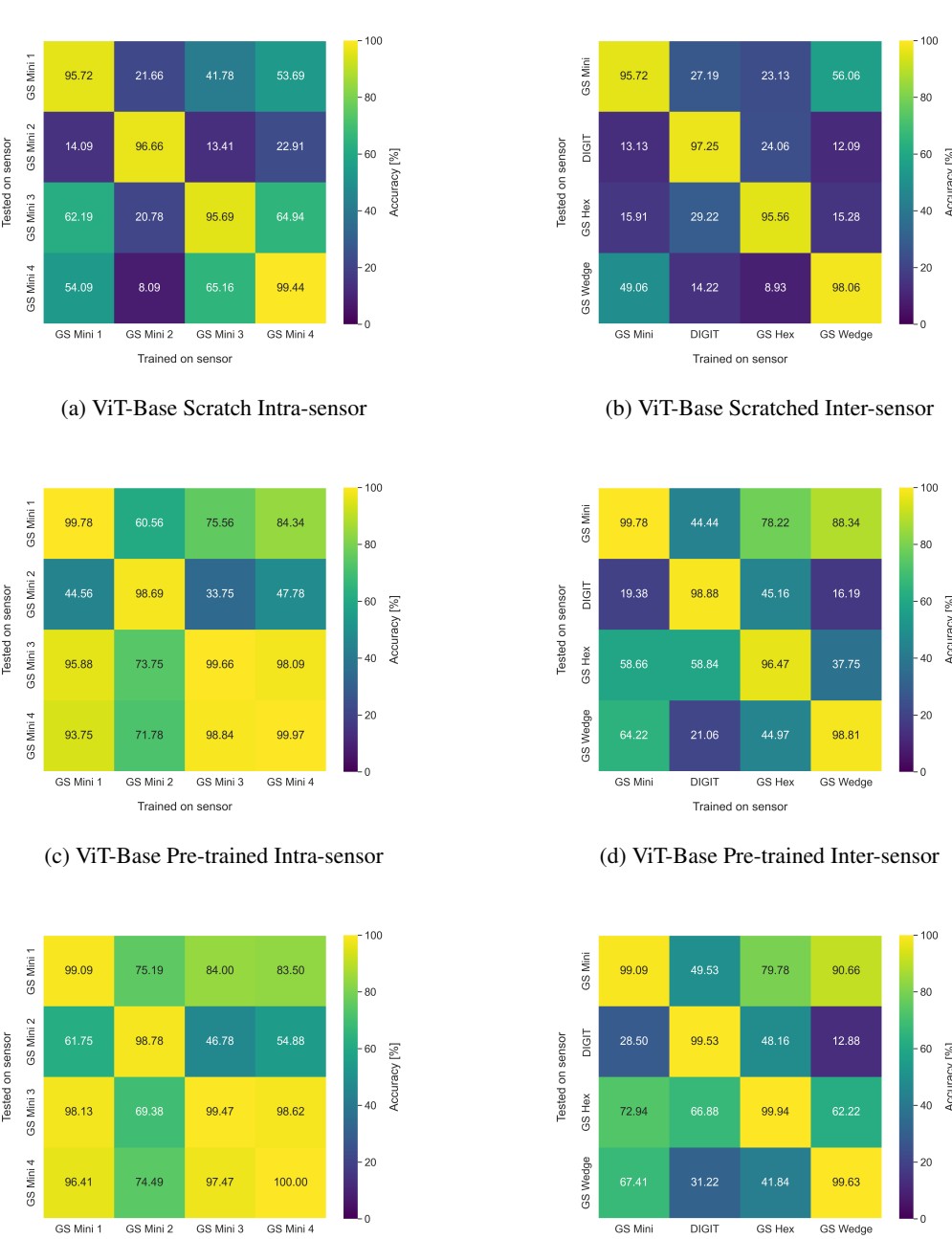

Figure 14: Transferability on classification tasks. (Part 1)

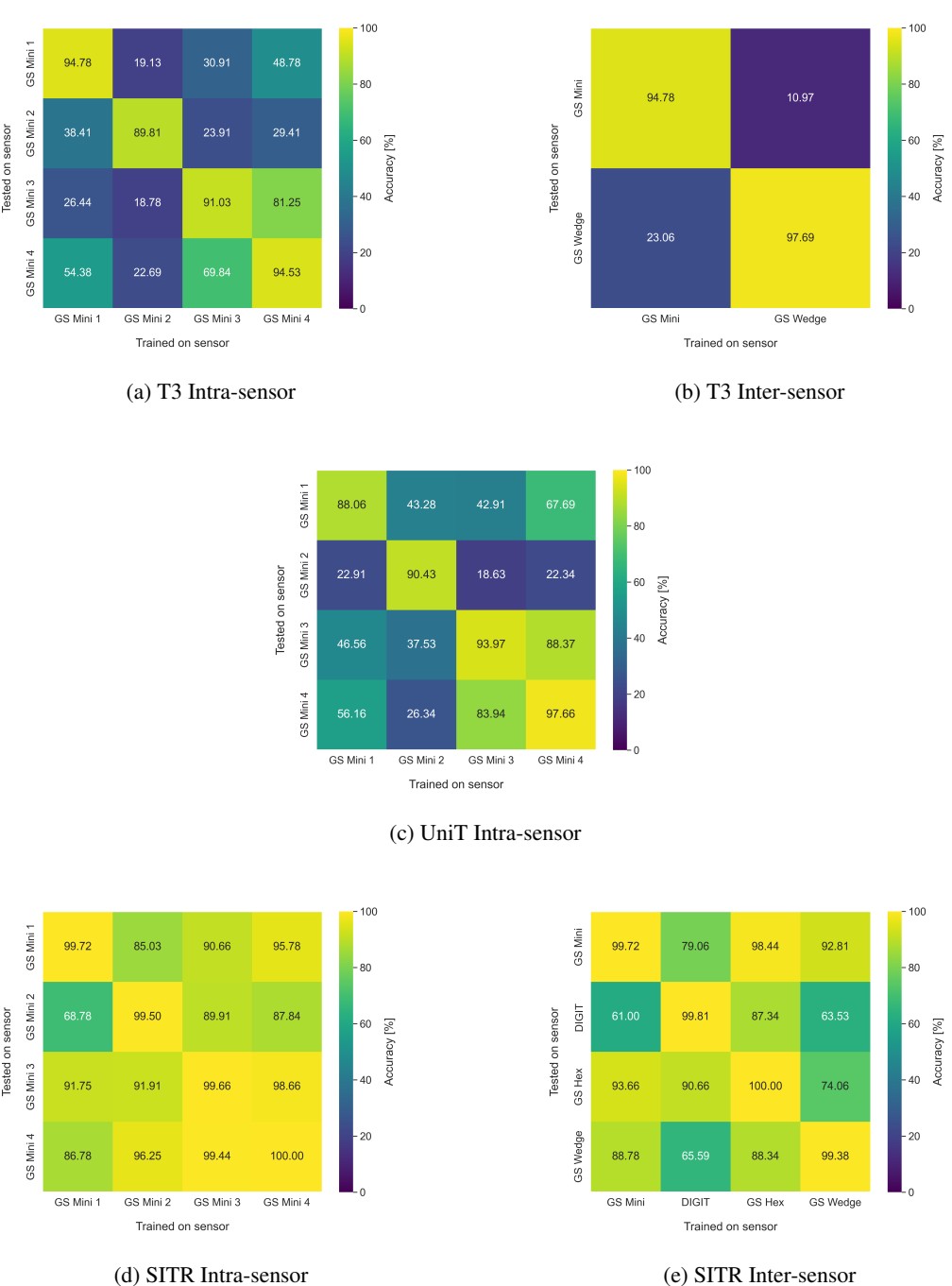

(a) T3 Intra-sensor

(b) T3 Inter-sensor

(c) UniT Intra-sensor

(d) SITR Intra-sensor

(e) SITR Inter-sensor

Figure 15: Transferability on classification tasks. (Part 2)

## A.5.2 POSE ESTIMATION

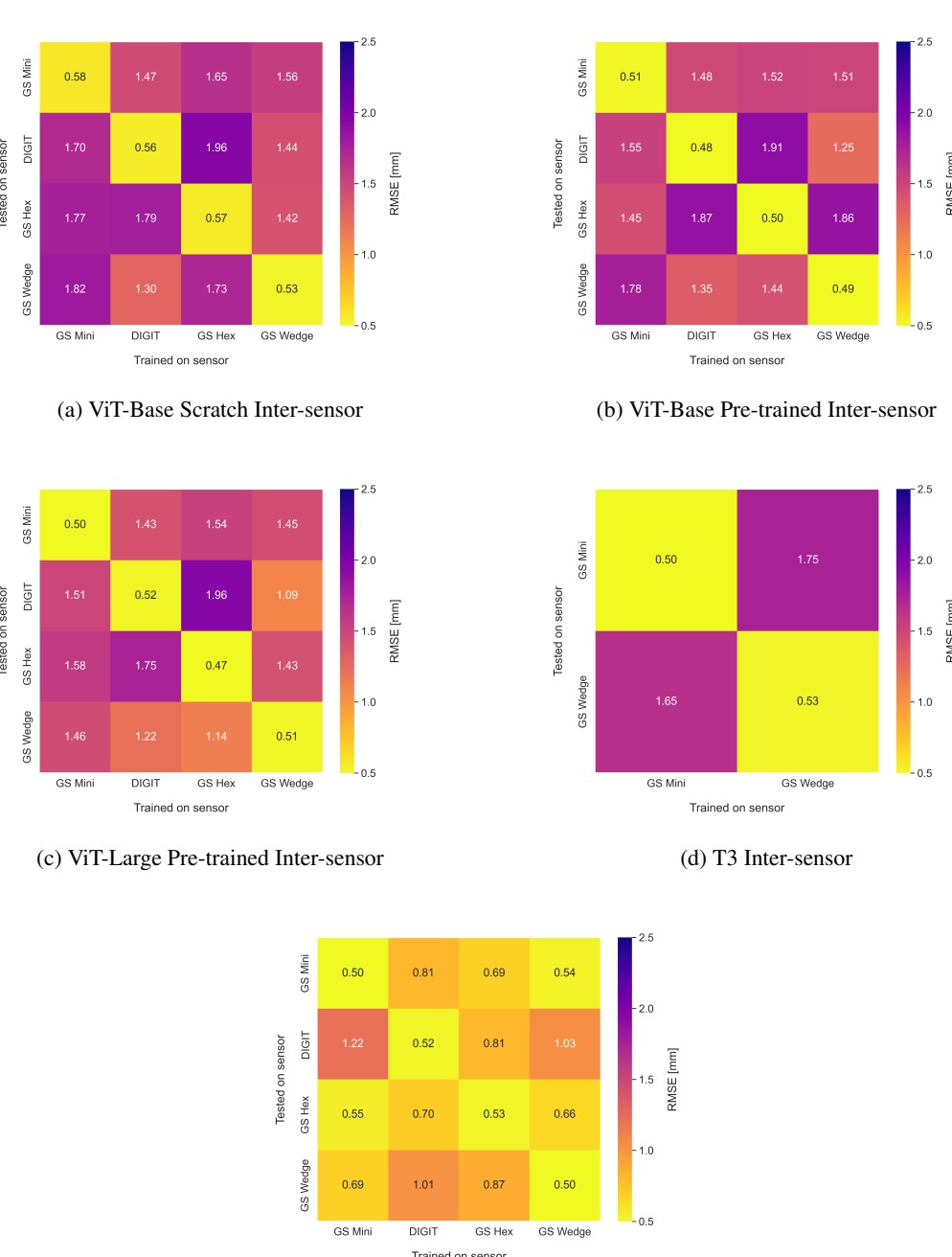

(a) ViT-Base Scratch Inter-sensor

(b) ViT-Base Pre-trained Inter-sensor

(c) ViT-Large Pre-trained Inter-sensor

(d) T3 Inter-sensor

(e) SITR Inter-sensor

Figure 16: Transferability on pose estimation tasks.

## A.6 Additional Ablations

This section presents ablation experiments to evaluate the impact of loss terms, alternative supervision signals, and dataset size on SITR's performance.

### A.6.1 Contribution of loss terms

We conduct an ablation study to evaluate the contributions of the normal map loss and SCL loss to SITR's performance. As shown in Table 5, either loss term independently serves as an effective supervision signal. However, their combination yields the strongest results. This evaluation is conducted on the dataset visualized in Figure 6, further highlighting how these two loss terms synergize to improve representation learning.

| Method | Classification (%) |
|---|---|
| Normal loss only | $84.21 \pm 14.01$ |
| SCL loss only | $78.86 \pm 18.72$ |
| Normal + SCL losses | $\mathbf{91.43} \pm 9.88$ |

Table 5: Ablation study showing the impact of different loss terms on classification accuracy transferability.

### A.6.2 Choice of supervision signal

There are alternative supervisions to our normal map, such as using MAE or VQGAN to reconstruct tactile images, as employed in T3 and UniT. To evaluate the effectiveness of SITR, we adapt these supervisions to train representations using our simulated dataset. We evaluate the models' transferability as described in Section 5.3 and Section 5.4. SITR consistently outperforms MAE and VQGAN, highlighting the benefits of SITR's architecture and training pipeline.

| Method | Classification (%) | | Pose estimation (mm) |
| | Intra-sensor set ↑ | Inter-sensor set ↑ | Inter-sensor set ↓ |
|---|---|---|---|
| MAE | $45.81 \pm 21.44$ | $26.46 \pm 19.54$ | $1.13 \pm 0.19$ |
| VQGAN | $59.41 \pm 19.50$ | $31.02 \pm 22.01$ | $1.18 \pm 0.14$ |
| SITR (Ours) | $\mathbf{90.23} \pm 8.16$ | $\mathbf{81.94} \pm 12.92$ | $\mathbf{0.80} \pm 0.21$ |

Table 6: Comparison of MAE, VQGAN, and SITR performance on intra-sensor and inter-sensor classification tasks (%) and inter-sensor pose estimation (mm)

### A.6.3 Effect of simulation dataset size

We evaluate how the size of the simulation dataset and the variety of sensor configurations impact classification transfer performance on inter-set classification. Table 7 shows that increasing the number of samples per sensor and the number of sensor variations lead to increases in performance. This demonstrates the benefit of a diverse and large-scale training dataset.

| Sensor Variations | Samples per sensor | | |
| | 1K | 5K | 10K |
|---|---|---|---|
| 10 | $45.82 \pm 21.12$ | $57.00 \pm 21.55$ | $61.44 \pm 22.81$ |
| 50 | $55.86 \pm 25.04$ | $68.55 \pm 11.96$ | $76.78 \pm 13.91$ |
| 100 | $62.85 \pm 16.45$ | $73.71 \pm 14.27$ | $\mathbf{81.94} \pm 12.92$ |

Table 7: Transfer classification accuracy (%) on the inter-set dataset across different sensor variations and samples per sensor.

