# OpenReview forum: "Sensor-Invariant Tactile Representation"
_ICLR.cc/2025/Conference — ICLR 2025 Poster_

### Official Review · Reviewer_cH4B · 2024-10-28

**Soundness:** 3
**Presentation:** 3
**Contribution:** 2
**Rating:** 6
**Confidence:** 5

**Summary:**

This work proposes a sensor-invariant tactile representation method that learns cross-sensor representations, enabling zero-shot generalization to new tactile sensors. To achieve this, the authors collect a large set of simulated tactile images paired with human-annotated images, where sensor contacts with the same objects in same pose are matched for positive pairs. Using a small amount of tactile data for downstream tasks, they fine-tune a task-specific head while keeping the pre-trained encoder frozen. They also demonstrate zero-shot generalization across unseen tactile sensors. The key contributions include: (i) an open-source large-scale dataset, (ii) zero-shot generalization to novel sensors, enabled by the model design, and (iii) a well-written, accessible paper.

**Strengths:**

The strengths of this paper are:

Originality: The problem is novel and relevant to practical applications of tactile-based learning.

Quality: he proposed method is well-founded, built upon established principles in the field, and presents a coherent theoretical framework. The experimental results, while only partially demonstrating the method's effectiveness, provide a solid initial validation of the approach. These findings indicate the potential for further development and optimization, suggesting that with additional refinement, the method could yield even more impactful results. However, a more comprehensive evaluation, including a broader range of experiments, would strengthen the evidence for the method’s effectiveness and applicability.

Clarify: The writing is clear and easy to follow.

Significance: Achieving zero-shot generalization to new tactile sensors is a valuable contribution to the field, addressing a critical challenge in tactile-based learning and manipulation. This capability has the potential to greatly expand the applicability of tactile sensing technologies across diverse robotic systems and tasks. By demonstrating that a model can effectively generalize to previously unseen sensors, the authors open up new avenues for research and development, ultimately advancing the state of the art in robotics and sensory integration. However, the tasks considered in this work is a bit limited.

**Weaknesses:**

There are several concerns:

Major:

1). The authors highlight the high cost of data collection, but recent studies (e.g., [1], [2]) have developed low-cost tools for gathering data across various manipulation tasks. I would like to see the authors' perspective on whether cross-sensor representation learning (SITR) remains necessary if data collection for each new sensor becomes straightforward. In other words, if sensor-specific data for any downstream task can be obtained easily, might single-policy approaches be more effective than a cross-sensor approach for certain tasks? Including a comparison in experiments could clarify this point, such as assessing whether the proposed method enhances sample efficiency in downstream task policy learning. This would help determine if easy access to sensor-specific data could potentially weaken the contribution of the cross-sensor approach in this work.

[1]. Chi, Cheng, et al. "Universal manipulation interface: In-the-wild robot teaching without in-the-wild robots." arXiv preprint arXiv:2402.10329 (2024).

[2]. Yu, Kelin, et al. "MimicTouch: Leveraging Multi-modal Human Tactile Demonstrations for Contact-rich Manipulation." 8th Annual Conference on Robot Learning.

2). For the SCL loss, collecting labeled data in simulation requires additional effort, as human experts must ensure correspondence in data collection (i.e., matching contact poses on the same object). Could advanced unsupervised methods, such as the approach in [3], be used instead to avoid direct label generation? My concern is that while such controlled data collection is feasible in simulation—where we can enforce contact poses and objects—it would be challenging in the real world, as replicating data collection across different tactile sensors is cumbersome.

[3]. Xu, Mengda, et al. "Xskill: Cross embodiment skill discovery." Conference on Robot Learning. PMLR, 2023.

3). The zero-shot experiments focus on estimation results, but how does the method perform in manipulation tasks that require control inference? Such tasks could be challenging, as the pre-training phase is limited to static simulated tactile images, lacking the temporal reasoning essential for dynamic manipulation tasks. Additionally, the authors generate 100 simulated sensor configurations, which may introduce bias from the rendered images during simulation pre-training. This setup could potentially increase the sim-to-real gap, particularly for complex manipulation tasks.

4). The dataset is extensive, with 1 million samples. It would be helpful to know if such a large dataset is necessary for achieving the reported results. An ablation study on dataset size could provide valuable insights into how dataset scale impacts performance.

5). The baseline evaluation may be somewhat unfair. It appears the authors used existing models for T3 and UniT, but these models may not have been trained on a dataset as large as the one used for the proposed method. If so, this could lead to a performance drop for the baseline models. Ensuring comparable training conditions would provide a more accurate assessment of model performance.

6). The calibration process partially reflects sensor characteristics. However, even without calibration images, the model’s performance remains significantly higher than T3, as shown in Table 1 and Figure 8. This raises concerns that the large-scale dataset and loss design may contribute more to performance than the calibration process or the model architecture itself. To better understand the actual impact of calibration and the proposed model architecture, it would be essential to train T3 models using the same dataset and loss.

7). In Figure 9, the performance without SCL loss is not substantially lower, even for classification tasks, though Figure 7’s t-SNE results show a visibly worse latent space without SCL. Could the authors clarify this discrepancy? Given that the SCL loss is central to this work, it’s surprising that in pose estimation tasks, performance without SCL is almost identical. This raises the question of whether SCL is as critical as suggested, since the reconstruction loss alone—commonly used in other works—seems sufficient to achieve cross-sensor learning if the model can reliably reproduce contact images. If so, this would suggest that the simulation data collection pipeline and use of contact IDs may not be as effective as proposed.

Minor:

8). Figure 2 is incorrect; it’s not the object ID that should be close in the latent space, but rather the contact ID, which encompasses both the object ID and a similar contact pose.

9). Is the learned decoder for reconstruction still utilized in downstream tasks? Since there is a fine-tuned task-specific decoder for each task, I assume the answer is no. It seems that the decoder used during pre-training is solely for the reconstruction loss to facilitate encoder training. Please clarify this in the paper.

**Questions:**

See weakness section. If my concerns are decently addressed, I am happy to increase the rating.

---

> ### Author Response · Authors · 2024-11-20
>
> Thank you for your detailed review and valuable feedback! We appreciate the time and effort you've dedicated to assessing our work, and we are grateful for your insights, which have helped us refine our contributions.
>
> **Cons and Questions:** We address your concerns and questions as follows:
>
>   - **Concern 1:**  *Is SITR necessary when data collection for new sensors becomes straightforward? If sensor-specific data for downstream tasks are cheap, will a single-policy approach be more effective?*
>
>     - **Response:** First of all, SITR is just a single-frame tactile representation. We appreciate your comments on using our methods in downstream robotic tasks like learning from demonstration which can be one of the good applications.
>       Even though data collection for each new sensor becomes straightforward, we'd mention that there are still differences between each individual sensor. That means when you release your trained model that uses tactile input, it suffers from the domain gap when being used on another device, even of the same sensor type. One of the main motivations of SITR is to address this inconvenience in tactile-informed robotic research. Compared to re-collecting downstream data, we believe pressing balls and cubes on sensors can lead to a brief yet effective standard for tactile-based robotic tasks.
>
>   - **Concern 2:**  *Controlled data collection is feasible in simulation, but challenging in the real world. Could advanced unsupervised methods be used to avoid that?*
>
>     - **Response:** We only need simulation to learn robust representations during SITR pre-training. This process aims to explore the latent space for various vision-based tactile sensor configurations, and we don’t need real-world datasets due to the variety of required sensors and objects.
>
>   - **Concern 3:**  *Can SITR perform well on robot manipulation tasks? Also, the 100 simulated sensor configurations may lead to bias and increase the sim2real gap.*
>
>     - **Response:** SITR can give a good transfer regardless of the nature of signals and, therefore can be fully integrated into any follow-up work regardless of either the dynamics of the robot or the tactile sensor used on it. Specifically, for manipulation tasks, we have demonstrated the potential of SITR to understand the contact geometry, which is essential for those tasks. We will demonstrate SITR on real robots with control in our follow-up work. As the purpose of SITR is to focus on the unification of sensor signals, the research on manipulation will be a standalone research question.
>       Regarding the sim2real gap, we showcased reasonable results on real sensors in our Experiments section. We base our simulation on the design principles of VBTS rather than modeling any single sensor. Particularly, DIGIT and GelSight Hex have significantly different optical designs than our simulation, and SITR shows strong transferability in our downstream experiments.
>
>   - **Concern 4:**  *Is a dataset of 1M samples necessary for pre-training?*
>
>     - **Response:** We conducted an ablation on this. As shown in Table 7 in our updated manuscript, the dataset size, and simulated sensor variance contribute a lot to SITR’s performance on classification transferability. We show SITR’s performance trained from a 10K dataset and up to a 1M dataset.
>
>   - **Concern 5:**  *The comparison between SITR and T3 may be unfair as they are not trained on such a large dataset.*
>
>     - **Response:** SITR is a combination of training architecture and dataset. As we claimed in the manuscript, the dataset is part of our contribution as well. Further, T3’s dataset includes over 3M samples. Despite that, we conducted ablation on T3 and UniT methods, shown in Table 6 in our updated manuscript. As we have different task settings and definitions of sensor types compared to T3, we only use their MAE supervision with our SITR pretraining. While T3 and UniT benefit from pretraining using our simulated dataset, SITR outperforms them across all settings.

---

> > ### Author Response · Authors · 2024-11-20
> >
> > - **Concern 6:**  *Without calibration, SITR is still performing better than T3. Does it indicate dataset is the main contributor to the performance?*
> >
> >     - **Response:** Yes, the dataset is indeed one of the main contributors to SITR's performance. Having a densely labeled dataset, which includes contact IDs and normal maps across various sensor configurations, allows us to train models that effectively capture both sensor-invariant and contact-specific features.  Also, please note that we claimed the dataset itself as part of our contribution. SITR is not only a learning framework but also contains the pipeline to render various vision-based tactile signals for pre-training.
> >
> >   - **Concern 7:**  *In Fig 9, the performance w/o SCL is not substantially lower but in Fig 7 it’s visibly worse. Can you clarify this discrepancy? Is SCL as critical as suggested?*
> >
> >     - **Response:** We show the inter-sensor classification results on the dataset used in Fig 7 in our updated manuscript as Table 5. Either normal loss or SCL loss can lead to a decent model on its own, while their combination leads to a boost from 84 to 91, matching our Fig. 9 results. As seen in Fig. 1, differences in gel thickness, camera parameters, and sensing scale cannot be bridged through reconstruction alone. Yet, as shown in Fig. 7, we can bridge those differences in the representation space and lead to better downstream performance. With our 2 losses, we hope to incorporate non-overlapping information signals into SITR that may be beneficial in various downstream tasks.
> >
> >   - **Concern 8:**
> >
> >     - **Response:** Fixed. Thank you for pointing this out!
> >
> >   - **Concern 9:**
> >
> >     - **Response:** Thank you for your suggestion. We clarified this in the caption of Fig. 2 in our latest revision.
> >
> > Once again, thank you for your valuable feedback. We hope our responses and the revised manuscript address your concerns effectively, and we welcome any further questions or comments that could help clarify our work.

---

> ### Comment · Reviewer_cH4B · 2024-11-25
> **Thank you for your detailed response**
>
> Thank you for your responses! My concerns regarding points 1–9 have been fully addressed. However, I have one additional question and would appreciate clarification from the authors.
>
> In [1], it is emphasized that properly handling in-batch data sampling is critical, as failing to do so could lead to the generation of easy negatives for contrastive learning, especially given the distinct characteristics of calibration images. Did you encounter a similar issue in your work? Additionally, did you employ any specific batch sampling strategies during pre-training or fine-tuning for downstream tasks?
>
> [1]. Yang, Fengyu, et al. "Binding touch to everything: Learning unified multimodal tactile representations." Proceedings of the IEEE/CVF Conference on Computer Vision and Pattern Recognition. 2024.

---

> > ### Author Response · Authors · 2024-11-29
> >
> > Binding Touch to Everything focuses on learning a representation from existing datasets, so it is essential to come up with the in-batch data sampling strategy to overcome the dataset imbalance. On the other hand, we collect our own real-world and simulation data for SITR training and testing for a full-fledged pipeline, thus guaranteeing the data balance.
> >
> > **Dataset Composition:**
> >
> > - Touch and Go dataset (GelSight Variation 1): 120K samples across 20 categories.
> > - The Feeling of Success dataset (GelSight Variation 2): 9.3K samples across 106 objects.
> > - YCB Slide dataset (DIGIT): 183K samples across 10 objects.
> > - Object Folder 2.0 dataset (Taxim): 180K samples across 1,000 objects.
> >
> > **Potential Sampling Issues:**
> >
> > Together, Touch and Go and YCB Slide account for approximately two-thirds of the training dataset, with no category overlap. I.e. the DIGIT sensor may never see a category seen by GelSight Variation 1.
> >
> > If batches are sampled disproportionately from specific datasets, the model might learn to associate objects with a sensor rather than the contact geometry on that sensor (i.e. easy negatives due to domain gap).
> >
> > For example, The Feeling of Success dataset makes up less than 2% of the training dataset. In a batch size of 48 (setting used in Binding Touch to Everything), it is likely to appear as a sole image-tactile pair per batch, making it challenging for the model to effectively learn representations from this dataset without oversampling during contrastive training.
> >
> > **Comparison to SITR:**
> >
> > In our work, the training dataset is evenly distributed across tactile sensors and contact geometries. This mitigates the risk of learning spurious associations between datasets and sensors and ensures a balanced contribution from all sensor domains during contrastive learning.
> >
> > We ensure that each batch sample has at least 1 positive example from another sensor variant. Beyond this, we do not modify our batch data sampling strategy.
> >
> > Thank you again for your feedback. If you have any further questions, please feel free to ask. We would be happy to address them.

---

> > > ### Comment · Reviewer_cH4B · 2024-11-30
> > > **Thank you for your detailed response**
> > >
> > > Thank you for your comment! I am willing to increase the score.

---

### Official Review · Reviewer_x9aX · 2024-11-01

**Soundness:** 3
**Presentation:** 3
**Contribution:** 2
**Rating:** 6
**Confidence:** 5

**Summary:**

This study introduces a novel approach for extracting Sensor-Invariant Tactile Representations (SITR) aimed at enhancing the transferability of models across different vision-based tactile sensors. The authors employ a transformer-based architecture combined with supervised contrastive learning to create a robust tactile representation that allows for zero-shot transfer across various optical tactile sensors, specifically within the GelSight series. Experimental results validate the approach’s effectiveness across multiple tactile sensing applications, indicating potential advancements in model generalization and transferability for similar sensor designs.

**Strengths:**

1. Originality and Innovation: The integration of a transformer-based architecture with supervised contrastive learning to achieve sensor-invariant tactile representation is a novel approach within the tactile sensing field. The method is original and demonstrates the potential for addressing the long-standing issue of sensor-specific dependency in tactile applications.
2. Technical Rigor and Methodological Soundness: The paper presents a technically robust framework, comprising a two-stage process of normal map reconstruction and contrastive learning. The experiments cover a range of scenarios, lending empirical support to the effectiveness of the proposed method. The methodology is well-designed to achieve sensor invariance, and the use of transformers in tactile representation learning is a valuable contribution.
3. Potential for Real-World Impact: Achieving zero-shot transfer with minimal calibration across sensors is valuable in robotics and other domains that involve tactile sensing. By minimizing the need for sensor-specific calibration, the proposed method has potential implications for resource efficiency and ease of deployment in tactile applications.

**Weaknesses:**

1. Limited Generalizability: The approach is specifically tailored to GelSight sensors, limiting its generalizability to other types of tactile sensors. GelSight sensors are vision-based, and the proposed method may not be easily transferable to non-vision-based tactile sensors, such as capacitive or resistive types. This restriction diminishes the broader applicability of the findings, potentially limiting its impact within the ICLR community.
2. Practical Constraints of GelSight Sensors: GelSight sensors, due to their relatively large size and offline nature, present challenges for real-time or embedded applications, especially in constrained environments such as robotic end-effectors. The physical limitations of GelSight sensors may limit the practical deployment of the method in real-world applications, where compactness and real-time processing are often required.
3. Insufficient Justification for Calibration Object Selection: The study uses a ball and cube corner for calibration, yet lacks a theoretical explanation for why these shapes are specifically suited for achieving sensor invariance. A more detailed discussion of the rationale and potential generalization of these calibration objects would improve the clarity of the methodology and support its robustness across varied sensor designs.
4. Parameter Tuning Transparency: The paper does not provide adequate information on the parameter tuning process, particularly for the temperature parameter in the contrastive learning phase. This parameter significantly influences the performance of contrastive learning, and a more transparent tuning discussion would enhance reproducibility and provide insight into the method’s robustness.
5. Computational Complexity and Real-World Feasibility: Transformer models, while powerful, are computationally intensive and may not be optimal for real-time applications. The authors do not discuss computational demands or potential limitations in deploying the model on resource-constrained platforms. An assessment of computational requirements would provide a more comprehensive view of the method’s practicality.

**Questions:**

1. Calibration Object Choice: Could you elaborate on the theoretical and practical reasoning for selecting a ball and cube corner as calibration objects? How do these specific shapes contribute to achieving sensor invariance, and would the method remain effective with other shapes?
2. Generalizability to Non-Vision-Based Sensors: Has the SITR approach been tested or considered for use with non-vision-based tactile sensors (e.g., capacitive or resistive)? If not, what challenges or limitations do you foresee in generalizing this approach to other sensor types?
3. Temperature Parameter in Contrastive Learning: Could you provide insights into how the temperature parameter was tuned in contrastive learning? This parameter plays a crucial role in the learning process, and understanding its tuning would aid in assessing the approach’s sensitivity and reproducibility. Additionally, given that GelSight sensors tend to generate considerable heat during use, with noticeable temperature changes within a minute, how might such temperature variations impact the performance or stability of your model?
4. Feasibility for Real-Time Applications: Given the computational intensity of transformer-based architectures, have you assessed the feasibility of this approach in real-time applications? Would it be possible to optimize the model for deployment on resource-constrained platforms?
5. Given that GelSight sensors are tested independently in your experiments rather than being integrated on a robotic platform (e.g., mounted on both sides of a gripper), could you discuss any potential modifications or limitations of your method for practical deployment in space-constrained environments, such as on robotic end-effectors?
Thank you for your thorough and thoughtful work in this area. The study is innovative and contributes valuable insights to the tactile sensing field. We look forward to your clarifications and insights, which I believe will further enrich the impact of this research and its relevance to the ICLR community！

---

> ### Author Response · Authors · 2024-11-20
>
> Thank you for your detailed review and valuable feedback! We appreciate the time and effort you've dedicated to assessing our work, and we are grateful for your insights, which have helped us refine our contributions.
>
> **Cons and Questions:** We address your concerns and questions as follows:
>
>   - **Concern 1:**  *SITR's approach is tailored to GelSight sensors and cannot transfer to non-vision-based sensors.*
>
>     - **Response:** SITR is trained and evaluated on vision-based tactile sensors. Please note we also evaluated SITR on non-GelSight sensors like DIGIT. For non-vision-based sensors, it is beyond the scope of this paper. As we mentioned in the Discussion section, we plan to extend SITR to non-vision-based sensors in our future research.
>
>   - **Concern 2:**  *GelSight sensors may not be a good choice for real-world applications due to their size and real-time performance*
>
>     - **Response:** SITR can be applied to various vision-based tactile sensors (VBTS), not limited to the original GelSight design. For example, GelSight Mini is a commercialized sensor suitable for robotic tasks, while other VBTS like Wedge embed the optical system into fingers, further addressing the size issue. Regarding the speed, GelSight Mini sensors can stream tactile images at 25 to 30 fps, while SITR can perform inference at ~200 fps, suitable for real-time applications.
>
>   - **Concern 3:** *Lacking a theoretical explanation of why balls and cubes are good calibration objects.*
>
>     - **Response:** As described in Sec 3.1, we chose the ball as a calibration object following a common practice. The intuition behind it is theoretically you can find all possible surface normal vectors on the ball to fit the RGB-to-gradient projection for VBTS. In addition to the ball, we tested the cube in SITR to inform the model of how the sensor surface deforms on object edges and corners.
>
>       A minor concern is that we use these two objects as they are easy to acquire, making the calibration step simple and reproducible. We believe other calibration objects will also work, as long as they contain similar surface normal information as these two.
>
>   - **Concern 4:** *The parameter tuning process is not transparent, especially for the contrastive learning temperature.*
>
>     - **Response:** We showed the results for different temperatures in Fig 9. Please let us know if you have questions regarding any other parameters.
>
>   - **Concern 5:** *Transformers are computationally intensive and may not be good for real-time applications. Please show the performance or limitations.*
>
>     - **Response:** We conducted experiments to test the inference speed. On RTX 4070 Ti Super, SITR can run at 178.25 fps, and 103.46 fps on RTX 2080 Ti, which is more than enough for real-time applications in real-world scenarios.
>
>   - **Question 1:** Same as concern 3.
>
>   - **Question 2:** Same as concern 1.
>
>   - **Question 3:** For the contrastive learning temperature we explained in response to concern 4; that the heat generated by GelSight sensors does not affect our experiment in both sensor speed and the captured tactile images.
>
>   - **Question 4:** Addressed in the response to concern 5.
>
>   - **Question 5:** Addressed in the response to concern 2. Also, grippers like WSG-50 have enough space for GelSight Mini sensors on both sides. No further modification is needed.
>
> Once again, thank you for your valuable feedback. We hope our responses and the revised manuscript address your concerns effectively, and we welcome any further questions or comments that could help clarify our work.

---

> ### Comment · Reviewer_x9aX · 2024-11-25
> **Clarification on Generalization Claims and Future Potential of SITR**
>
> Thank you for your detailed responses to my comments. Your clarifications address several of the concerns raised, and I appreciate the effort you have put into refining your work. Below, I provide follow-up thoughts on your responses and additional suggestions for improving the manuscript:
>
> Concern 1: SITR's approach is tailored to GelSight sensors and cannot transfer to non-vision-based tactile sensors.
> I understand that SITR focuses on vision-based tactile sensors, and I appreciate your clarification that extending to non-vision-based sensors is beyond the scope of this paper. However, as DIGIT is based on GelSight technology, it should not be classified as a "non-GelSight sensor." DIGIT is part of the same vision-based tactile sensor ecosystem, which limits the scope of your generalization claim.
> Suggestions:
> To strengthen the claim of sensor invariance, I recommend including a discussion on the potential challenges and methodologies for extending SITR to non-vision-based tactile sensors in future work. This would broaden the impact and relevance of your study.
>
> Concern 4: The parameter tuning process is not transparent, especially for the contrastive learning temperature.
> Thank you for referencing Fig. 9 for the tuning results. However, it would be helpful to include more details on how the final temperature value was selected (e.g., empirical testing, cross-validation) and to elaborate on SITR’s sensitivity to this parameter.
> Suggestions:
> 1) Provide a detailed explanation of the parameter tuning methodology and its impact on downstream tasks to improve transparency.
> 2) Include results showing performance variation across a range of temperature values to demonstrate the robustness of SITR to parameter changes.
>
> Your responses have clarified many points and demonstrate the significant contributions of SITR to tactile sensing research. Addressing the above suggestions in the revised manuscript would further strengthen its rigor and impact. If these areas can be addressed or further clarified, I am willing to raise my score to reflect the improved quality and comprehensiveness of your work.
>
> Thank you again for your thoughtful replies, and I look forward to seeing the updated version of your submission.

---

> > ### Author Response · Authors · 2024-11-29
> >
> > Thank you for your detailed review and thoughtful comments.
> >
> > Regarding your concern 1: Thanks for your suggestions. We have extended our discussion section to include potential challenges and our preliminary thoughts on potential key elements required for extending SITR to non-vision-based sensors. We leave the actual implementation to our future work.
> >
> > Regarding your concern 4: We have revised Fig. 9 to include results with more comprehensive temperature settings, which now shows a comparison of SITR’s performance on classification and pose estimation across 6 different temperature settings. As shown, t=0.07 has the best overall performance across three tasks. We have therefore empirically chosen this value for our experiments.
> >
> > Thank you again for your feedback. If you have any further questions, please feel free to ask. We would be happy to address them.

---

### Official Review · Reviewer_rhaT · 2024-11-03

**Soundness:** 3
**Presentation:** 3
**Contribution:** 3
**Rating:** 6
**Confidence:** 4

**Summary:**

To address the problem of current vision-based tactile sensors' lack of transferability, this paper proposes a transformer-based method called SITR. SITR is trained on simulated tactile data, and is generalizable to real-world sensors though calibration. Experiments show the zero-shot performance on three downstream tasks: shape reconstruction, object classification and contact localization.

**Strengths:**

1. Sensor variance, which is the problem this paper tries to solve, is very important in the field of vision-based tactile sensing. Recently, different methods [1, 2, 3] have been proposed to address this problem, and this work proposes a model that is complementary to the prior works.

2. The framework proposed by this paper does not require any sensor-specific design in the model architecture, which seems to be its major contribution. Instead of designing different encoders or tokens for specific sensors, SITR only relies on calibration images to distinguish between different sensors. This can both reduce the model size and enable transferability to entirely novel sensors (as long as calibration images can be obtained).

3. The zero-shot results on downstream tasks show that the model transfers well across various sensor settings, which is a significant improvement upon the prior works.

References
[1] Zhao, Jialiang, Yuxiang Ma, Lirui Wang, and Edward H. Adelson. "Transferable Tactile Transformers for Representation Learning Across Diverse Sensors and Tasks." arXiv preprint arXiv:2406.13640 (2024).
[2] Rodriguez, Samanta, Yiming Dou, Miquel Oller, Andrew Owens, and Nima Fazeli. "Touch2touch: Cross-modal tactile generation for object manipulation." arXiv preprint arXiv:2409.08269 (2024).
[3] Higuera, Carolina, Akash Sharma, Chaithanya Krishna Bodduluri, Taosha Fan, Patrick Lancaster, Mrinal Kalakrishnan, Michael Kaess et al. "Sparsh: Self-supervised touch representations for vision-based tactile sensing." In 8th Annual Conference on Robot Learning.

**Weaknesses:**

1. The dataset the model is trained on all comes from simulation. The tactile simulator uses 3D meshes of the object to synthesize RGB tactile images, which assumes that all objects are rigid and without micro-geometry on the object surface (e.g., wood grain). Although this might be sufficient for geometry-based tasks such as shape reconstruction and object classification, the features extracted from this model may lack the ability to classify the fine-grained details (e.g., smoothness, hardness) in real-world setting. See the Questions section for more suggestions.

2. Despite being a sensor-agnostic model architecture, the calibration images are necessary for the SITR model. It's also shown in the ablation study that the model performance will drastically drop when calibration images are removed, and more calibration images usually lead to better performance. This leads to two potential concerns: (i) the calibration of new sensors requires specific objects (ball and cude) that might not be easily available to some of the users, which limits the application of the proposed model; (ii) the calibration images can lead to significant computation overhead, and most of the regions in the calibration images are redundant, i.e., not contacted by the object (shown in Fig. 3). See the Questions section for more suggestions.

**Questions:**

1. Referring to my first concern in Weaknesses section, it would be interesting to test the model's transferability to real-world datasets with more complicated settings. For example, Touch and Go [1] proposes three material understanding tasks that evaluate the model's understanding of material categories/ smoothness/ hardness given GelSight images. It would be interesting to see if the features learnt from pure simulation data can be transferred to these tasks.

2. Referring to my second concern, I'm curious about the computation overhead caused by using 18 calibration images. Also I wonder if a more careful design can be used in encoding the calibration images to reduce redundant information. As is shown in Fig. 3, only one out of the nine grids contains useful information. One simple potential improvement would be cropping the useful regions out from the 9 calibration images and concatenating them into a single image.

References
[1] Yang, Fengyu, Chenyang Ma, Jiacheng Zhang, Jing Zhu, Wenzhen Yuan, and Andrew Owens. "Touch and go: Learning from human-collected vision and touch." arXiv preprint arXiv:2211.12498 (2022).

---

> ### Author Response · Authors · 2024-11-20
>
> Thank you for your detailed review and valuable feedback! We appreciate the time and effort you've dedicated to assessing our work, and we are grateful for your insights, which have helped us refine our contributions.
>
> **Pros:** Both the motivation and technical approach ideas come from our practical use of tactile sensors. We appreciate your recognition of the importance of achieving sensor-invariance and our corresponding design towards it.
>
> **Cons and Questions:** We address your concerns and questions as follows:
>
>   - **Concern 1:**  *As SITR is trained on simulated 3D datasets, can it generalize to downstream tasks that require fine-grained details like smoothness and hardness in real-world settings? For example, Touch-and-Go dataset?*
>
>     - **Response:** We appreciate your notice on more downstream tasks in tactile perception. As Touch-and-Go and other previous tactile datasets don't have calibration images for the sensors, we cannot directly do experiments on those datasets with SITR. Instead, we'd like to clarify that the fine-grained details you mention, including smoothness and hardness, are predicted based on the dynamics of contact geometry in those datasets. Currently, SITR is designed for single-frame representations, but it can be integrated into existing prediction frameworks for such tasks given calibration images.
>
>       We show several surface geometry reconstruction examples in our supplementary material on real-world objects to show our extracted features are capable of preserving fine-grained details even for deformable objects with textures and, therefore have the potential to be applied to such scenarios.
>
>   - **Concern 2:**  *Regarding calibration steps, (i) specific calibration objects are not easily available, and (ii) some calibration images might be redundant -- will cropping and concatenation improve the performance?*
>
>     - **Response:** We appreciate your concerns and suggestions on our use of calibration images. Regarding your concern (i), the reason we use these two objects is, that the ball indentor covers all possible surface normals for calibration, while the corner of a cube can provide more information on how gel pads deform on edges and corners. A ball of a certain diameter is not hard to acquire in most labs, while for the cube you can use any object with a corner to replace it.
>
>       For (ii), As mentioned in section 3.1, for calibration token pre-processing, we concatenate the calibration images along the channel dimension and project them into tokens before feeding them into the transformer encoder. That means we always have the same number of tokens regardless of the number of calibration images. Here, the number of calibration tokens is only dependent on the patch size (default used in ViT).
>
>       We conducted ablation on the number of calibration images. For models without calibration images, SITR runs at 203 fps on RTX 4070 Ti Super, while dropping to 178 fps with any number of calibration images. Cropping and concatenating calibration images into one results in no difference in inference speed.
>
>   - **Question 1:** Same as Concern 1.
>
>   - **Question 2:** Same as Concern 2.
>
> Once again, thank you for your valuable feedback. We hope our responses and the revised manuscript address your concerns effectively, and we welcome any further questions or comments that could help clarify our work.

---

> > ### Comment · Reviewer_rhaT · 2024-11-26
> >
> > Thanks for the detailed response!
> > Regarding my two concerns, I think the second one is mostly solved by the provided inference speed.
> > However my first concern on using simulated datasets still holds. It seems that SITR can only be trained on the datasets with calibration images, which are not included in most of the real-world datasets. This would largely limit the model's capability of transferring to real-world settings. The provided experiment results are also not showing that the model can extract fine-grained features from real-world data.
> > Therefore I'll keep my rating and vote towards marginally above acceptance.

---

> ### Author Response · Authors · 2024-11-29
>
> Thank you for your detailed review and thoughtful comments.
>
> We recognize that most publicly available tactile datasets lack calibration images, which limits SITR's direct applicability. However, calibration is a common practice in tactile sensing research and proves especially beneficial for inter-sensor consistency. We hope that SITR's demonstrated advantages will encourage future dataset releases to include such calibration data, enabling more robust generalization in the field.
>
> Regarding the additional results, we hope to show that fine-grained geometry from real-world data is what downstream tasks for hardness/smoothness/etc prediction rely on. Due to the dataset and time constraints, we do not show those downstream results at this time. Even though, we believe a representation containing high-resolution geometry information will be sufficient for those downstream tasks.
>
> Thank you again for your insights and for helping us refine this work. We remain open to further questions or suggestions.

---

### Official Review · Reviewer_3T4J · 2024-11-05

**Soundness:** 3
**Presentation:** 3
**Contribution:** 3
**Rating:** 6
**Confidence:** 4

**Summary:**

This paper aims to learn a tactile representation that can be used across different tactile sensors, including the same sensor design but with different instances, or different sensor designs. To achieve this, it collects a large dataset in simulation and uses contrastive learning to learn the representation. It divides a tactile image into patches and encodes it using a transformer. Meanwhile, it also passes simple calibration images, which contain simple objects pressing on different corners, to this transformer. The feature is further refined by learning to predict the normal map. The learned representation can be fine-tuned on downstream tasks and significantly outperforms baselines and other state-of-the-art methods.

**Strengths:**

The idea of using simple calibration images is interesting and inspiring. I believe this design should become a standard technique in future tactile representation learning.

The performance of the learned representation is quite good. On the three downstream tasks they evaluate, it significantly outperforms baselines and other methods in this area.

It also presents comprehensive ablation experiments on the role of calibration images and contrastive learning losses.

The paper provides nice visualizations of the learned feature space.

**Weaknesses:**

I believe there should be another important ablation experiment: training UniT/T3 on the simulated images collected in this paper. This is because there are two major differences between previous state-of-the-art methods (UniT/T3) and the proposed method. The first difference is the specific architectural design, such as contrastive learning, the transformer, and the use of calibration images. The second is the use of simulation data versus real-world data. This paper already presents experiments on architectural design but does not show whether the good performance mainly comes from a large simulation dataset. Specifically, I think the authors could train a representation with the loss proposed in UniT/T3 and compare its performance with the proposed method. This would provide more insight into where the performance improvement comes from.

It seems the authors do not compare the performance gain achieved by predicting the normal map.

Another weakness of this work is that it only evaluates sensors with flat gel pads, which have very similar optical designs. In contrast, other work, such as T3, also includes curved sensors. This point should be clearly discussed in the paper.

**Questions:**

What is the performance of UniT or T3 when trained on the simulation dataset gathered in this paper?

What is the effect of using the normal map prediction loss?

Is the statement “increasing the number of calibration images does not incur additional inference costs, as calibration tokens are computed only once per sensor” correct? Although it does not introduce additional computational cost for tokenization, the self-attention operator cost is O(n^2), where n is the sequence length. If you increase the number of calibration images, that cost will grow quadratically.

How is the “No SCL” baseline trained? Is it trained only with the normal prediction loss?

---

> ### Author Response · Authors · 2024-11-20
>
> Thank you for your detailed review and valuable feedback! We appreciate the time and effort you've dedicated to assessing our work, and we are grateful for your insights, which have helped us refine our contributions.
>
> **Pros:** We are pleased that you found the motivation, experimental approach, and visualizations compelling. Developing a standardized technique for tactile-related learning tasks is indeed our primary goal, and we appreciate that these aspects resonated with you.
>
> **Cons and Questions:** We address your concerns and questions as follows:
>
>   - **Concern 1:**  *UniT and T3 differ from SITR in both architecture and dataset. Is it possible that SITR's performance gain comes from the dataset rather than architecture?*
>
>     - **Response:** SITR is a combination of training architecture and dataset. As we claimed in the manuscript, the dataset is part of our contribution as well. Despite that, we conducted ablation on T3 and UniT methods. As we have different task settings and definitions of sensor types compared to T3, we only use their MAE supervision in pretraining. While T3 and UniT benefit from pretraining using our simulated dataset, SITR outperforms them across all settings. For detailed results, please refer to Table 6 in our updated manuscript.
>
>   - **Concern 2:**  *Comparing the performance gain achieved by predicting the normal map.*
>
>     - **Response:** Predicting the normal map is a fundamental step in our architecture, working as the decoder and reconstruction loss in auto-encoders. It helps SITR preserve dense geometrical features for downstream tasks. VQGAN/MAE are forced to learn sensor-variant features by supervising on image reconstruction. We show demonstrable performance gains over those methods.
>
>         If you mean that we can train SITR with SCL loss only, then we did a transfer ablation shown in Table 5 of our updated manuscript. SITR will suffer a performance drop if either of them is missing.
>
>         If you mean why don't we predict a depth map instead of a normal map, it is due to the nature of optical-based tactile sensors. As shown in Sec. 3.1, normal vectors are directly connected to each RGB value, while depth is later integrated which may cause cumulative errors. In our preliminary experiments, models predicting depth maps perform worse than normal map versions.
>
>   - **Concern 3:** Evaluated only flat gel pads.
>
>     - **Response:** In our pre-training simulation, we rendered various configurations based on flat sensor models only. The current SITR model may not perform well on curved sensors without pre-training due to their slight difference in calibration and normal map prediction. We mention in our Discussion section that one of our future directions is to have curved sensors in our framework.
>
>   - **Question 1:** Same as Concern 1.
>
>   - **Question 2:** Same as Concern 2.
>
>   - **Question 3:**  *Will computation cost grow as we have more calibration images?*
>
>     - **Response:** As mentioned in section 3.1, for calibration token pre-processing, we concatenate the calibration images along the channel dimension, and project them into tokens before feeding them into the transformer encoder. That means we always have the same number of tokens regardless of the number of calibration images. Here, the number of calibration tokens is only dependent on the patch size (default used in ViT).
>
>   - **Question 4:** How is the "No SCL" baseline trained? Only with normal prediction loss?
>
>     - **Response:** Yes, we have clarified that in our updated manuscript in Sec 6.2.
>
> Once again, thank you for your valuable feedback. We hope our responses and the revised manuscript address your concerns effectively, and we welcome any further questions or comments that could help clarify our work.

---

### Meta-Review · Area_Chair_wGZQ · 2024-12-21

**Metareview:**

This paper introduces SITR (Sensor-Invariant Tactile Representation), a framework for learning a unified representation that transfers across different vision-based tactile sensors. The method combines a transformer-based architecture with supervised contrastive learning and calibration images to enable zero-shot transfer between sensors.

Main strengths identified by the reviewers (particularly focusing on the detailed feedback from Reviewers 3T4J and rhaT):
- A novel solution addressing the critical challenge of sensor variance in tactile sensing
- A method that effectively integrates calibration images with neural architectures
- Comprehensive empirical validation demonstrating strong zero-shot transfer capabilities
- Clear presentation and thorough experimental analysis

Limitations:
- Current evaluation focuses primarily on flat gel pad sensors
- Additional validation needed for downstream manipulation tasks
- As noted by all reviewers, the methods inherently rely on simulation data for training, which may introduce biases to the learned representation.

Justification for rating: The paper presents a decent contribution to tactile sensing with strong technical merit and practical impact. The experimental validation and the improvements over existing methods demonstrate its value to the field. While there are some limitations in scope, the core method is sound and well-validated. The authors have comprehensively addressed reviewer concerns and validated the method across multiple tasks and sensor types.

**Additional Comments On Reviewer Discussion:**

Key discussion points
- The authors demonstrated through ablation studies that SITR's improvement comes from both the architectural design and the dataset, not merely from dataset scale
- Initial concerns about the assumption made by the calibration step were addressed.
- Questions about real-world applicability were resolved: the authors demonstrated practical inference speeds (~200 fps) and successful transfer to commercial sensors
- Technical questions about contrastive learning parameters and batch sampling strategies were addressed with additional experimental results

The authors have been highly responsive throughout the discussion period, providing detailed clarifications and additional experiments that address the core concerns raised by the reviewers.

I would like to give a special highlight to reviewer cH4B's extensive comments and quick response.

---

### Decision · Program_Chairs · 2025-01-22

Accept (Poster)